# Opioid analgesia alters corticospinal coupling along the descending pain system in healthy participants

**Alexandra Tinnermann[1,2]\*, Christian Sprenger[1†‡], Christian Büchel[1,2†]**

[1]Department for Systems Neuroscience, University Medical Center Hamburg-Eppendorf, Hamburg, Germany; [2]Max Planck School of Cognition, Leipzig, Germany

**Abstract** Opioids are potent analgesic drugs with widespread cortical, subcortical, and spinal targets. In particular, the central pain system comprising ascending and descending pain pathways has high opioid receptor densities and is thus crucial for opioid analgesia. Here, we investigated the effects of the opioid remifentanil in a large sample (n = 78) of healthy male participants using combined corticospinal functional MRI. This approach offers the possibility to measure BOLD responses simultaneously in the brain and spinal cord, allowing us to investigate the role of cortico-spinal coupling in opioid analgesia. Our data show that opioids altered activity in regions involved in pain processing such as somatosensory regions, including the spinal cord and pain modulation such as prefrontal regions. Moreover, coupling strength along the descending pain system, that is, between the anterior cingulate cortex, periaqueductal gray, and spinal cord, was stronger in participants who reported stronger analgesia during opioid treatment while participants that received saline showed reduced coupling when experiencing less pain. These results indicate that coupling along the descending pain pathway is a potential mechanism of opioid analgesia and can differentiate between opioid analgesia and unspecific reductions in pain such as habituation.

**\*For correspondence:**
a.tinnermann@uke.de

[†]These authors contributed equally to this work

**Present address:** [‡]Center for Anesthesiology and Intensive Care Medicine, University Medical Center Hamburg-Eppendorf, Hamburg, Germany

## Editor's evaluation

Here, the authors used sophisticated methods for combined brain and spinal cord functional MRI. They report on the influence of an intravenous opioid, remifentanil (a potential, very short-acting μ-opioid receptor agonist), on ascending and descending pain processing pathways in healthy subjects. Their detailed analysis strengthens findings from previous human and animal studies and revealed novel changes in connectivity in the descending pathway to the spinal cord.

## Introduction

Opioids such as μ-opioid receptor agonists are potent analgesic agents and play a crucial role in pain therapy. Their targets, μ-opioid receptors, are expressed in large parts of the central pain system, including the ascending and descending pain pathway (*Corder et al., 2018*). The descending pain pathway and its role in opioid analgesia have been extensively studied in animals and to some extent in humans. One of the key regions of the descending pain pathway is the periaqueductal gray (PAG), which exerts top-down control onto the spinal cord dorsal horn through the rostral ventromedial medulla (RVM) (*Fields, 2004*). This control is bidirectional, resulting in a descending modulation that can either inhibit or facilitate spinal nociceptive processing. Experimental manipulations of signaling between the PAG, RVM, and spinal cord such as electrical stimulation of the PAG have been shown to strongly inhibit spinal nociceptive signals and induce profound analgesia in animals (*Liebeskind et al., 1973*; *Mayer and Price, 1976*) and humans (*Baskin et al., 1986*). Other experimental

manipulations involved opioid injections into the PAG and RVM that similarly led to profound analgesia (*Yaksh et al., 1988*). These findings suggest that the descending pain pathway is involved in analgesia and that opioids are an important neurotransmitter within this system. Moreover, the elimination of descending signaling between the RVM and the spinal cord attenuates the analgesic effects of systemic opioids in rodents (*Basbaum et al., 1977*; *Kiefel et al., 1993*), which further shows that one important mechanism of opioid analgesia is spinal inhibition through the descending pain pathway. Within the spinal cord, high μ-opioid receptor densities are mainly found in superficial layers (*Faull and Villiger, 1987*) but also to a lesser extent in deeper layers (*Wang et al., 2018*). Similarly to the PAG and RVM, opioid injections into the spinal cord result in profound analgesia (*Onofrio and Yaksh, 1990*; *Yaksh and Rudy, 1976*) and injections of an opioid antagonist into the spinal cord can block opioid analgesia through both opioid injection into single-brain regions (e.g., PAG) and systemic opioid administration (*Chen and Pan, 2006*; *Yaksh and Rudy, 1976*). These results suggest that opioids in the spinal cord can completely inhibit ascending nociceptive signals. Many opioid receptor-rich regions, including the spinal cord and PAG, integrate ascending nociceptive information with descending modulatory inputs (*Fields, 2004*). It is therefore reasonable to assume that opioid analgesia results from at least two, probably related, mechanisms, namely, reduced ascending pain signaling together with increased descending inhibition. In line with this hypothesis, neuroimaging studies in humans have shown that brain regions targeted by the major ascending pain pathway, that is, the spinothalamic tract, such as the insula, anterior cingulate cortex, thalamus, and somatosensory cortex, show that activity is consistently decreased during opioid administration (*Atlas et al., 2012*; *Bingel et al., 2011*; *Hansen et al., 2015*; *Wise et al., 2002*). A dose–response relationship was further observed in brain regions associated with pain intensity encoding such as the somatosensory cortex and posterior insula while activity in the anterior insula and amygdala was abolished at the lowest opioid dose (*Oertel et al., 2008*). Evidence for the involvement of the descending pain pathway in opioid analgesia comes from imaging studies that found opioid-related activation changes in the PAG and RVM that correlated with perceived opioid analgesia (*Wanigasekera et al., 2012*). Activity increases during opioid treatment have also been reported in the prefrontal cortex and PAG (*Wagner et al., 2007*), and coupling between those regions was enhanced during opioid treatment (*Petrovic et al., 2002*). Interestingly, enhanced coupling between the prefrontal cortex and PAG has also been described in placebo hypoalgesia (*Bingel et al., 2006*; *Eippert et al., 2009a*; *Ellingsen et al., 2013*; *Petrovic et al., 2002*; *Wager et al., 2007*), suggesting that opioid and placebo analgesia share a common coupling pattern within the descending pain system. This hypothesis is further supported by the fact that this placebo-related coupling pattern was abolished by an opioid antagonist (naloxone), which indicates that this mechanism depends on endogenous opioids (*Eippert et al., 2009a*). Apart from altered coupling, expectations such as placebo analgesia have been shown to modulate activity at the level of the spinal cord (*Eippert et al., 2009b*; *Geuter and Büchel, 2013*; *Matre et al., 2006*; *Tinnermann et al., 2017*), which further indicates that the coupling between the prefrontal cortex and PAG ultimately targets the spinal cord to modulate spinal processing. It is thus reasonable to hypothesize that systemic opioid application recruits the descending pain pathway similar to animal studies and further modulates coupling between regions of this pathway. Consequently, it is necessary to move beyond mere activations in individual areas of the central pain system and focus on the interactions between regions to better understand pain and opioid analgesia dynamics (*Kucyi and Davis, 2017*; *Kucyi and Davis, 2015*; *Lee et al., 2021*; *Upadhyay et al., 2010*). Recent advances in neuroimaging techniques have paved the way to model large-scale network interactions, including the brain, brainstem, and spinal cord (*Sprenger et al., 2015*; *Tinnermann et al., 2017*). We therefore combined a corticospinal functional MRI (fMRI) approach that allows measuring BOLD responses simultaneously in the brain and spinal cord (*Finsterbusch et al., 2013*) with a fast-acting intravenous μ-opioid receptor agonist (remifentanil) treatment. This allowed us to investigate the mechanisms underlying opioid analgesia within most regions of the central pain system and in particular to estimate the effect of remifentanil on coupling between the brain, brainstem, and spinal cord. We further included an expectation manipulation within the study design to test whether different levels of certainty (50% vs. 100% certainty) to receive remifentanil can modulate perceived analgesia, thereby expanding on previous studies that were able to show that the knowledge about receiving a potent analgesic drug enhanced perceived analgesia (*Atlas et al., 2012*; *Bingel et al., 2011*).

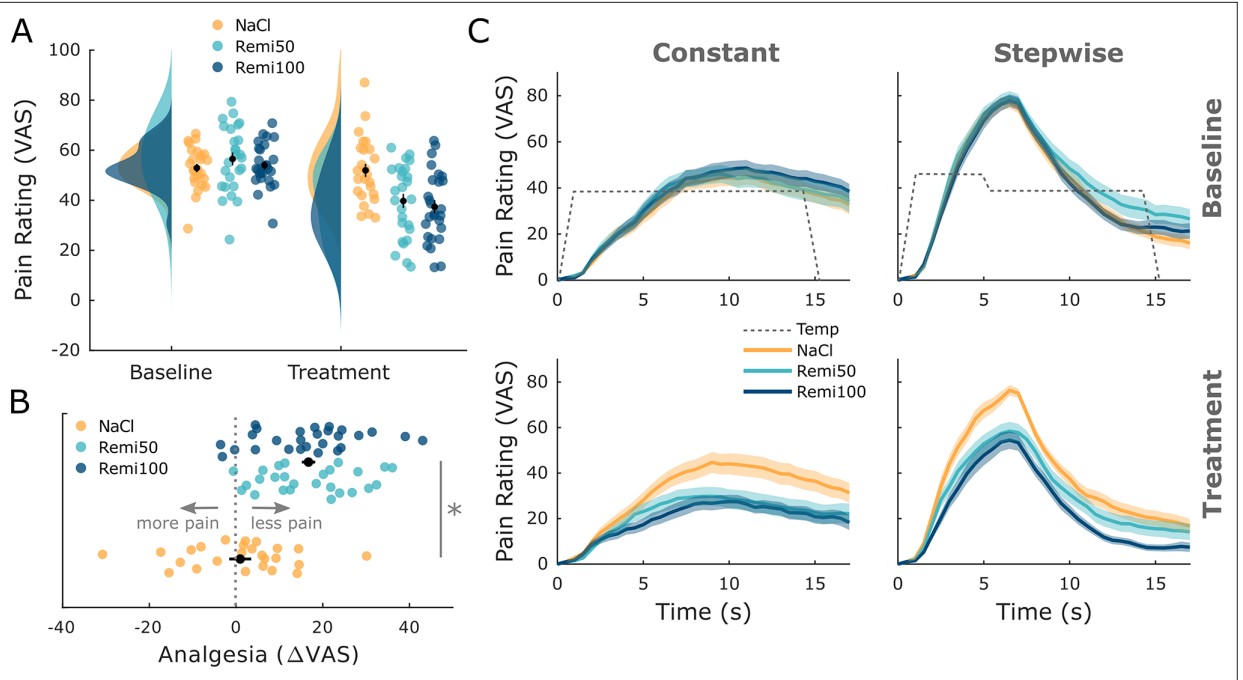

**Figure 1.** Behavioral results. (**A**) Pooled retrospective pain ratings across stimuli (constant, stepwise) and time points (T1, T2) show a significant pain reduction during opioid treatment but no significant difference between Remi50 and Remi100. (**B**) The difference between baseline and treatment ratings shows that a majority experienced analgesia during the remifentanil treatment. The labels 'less pain' and 'more pain' refer to the treatment phase compared to baseline. (**C**) Online pain ratings show a similar pattern: remifentanil reduced pain ratings for both stimuli while there was no significant difference between the Remi50 and Remi100 groups. Results were considered significant at $p_{corr}<0.05$. Error bars/shaded areas represent standard error of the mean.

The online version of this article includes the following figure supplement(s) for figure 1:

**Figure supplement 1.** Detailed behavioral results for all rating types.

**Figure supplement 2.** Functional MRI (fMRI)-related plots for all three experimental groups.

## Results

### Opioid effects on pain ratings

In the first analysis, we were interested in whether remifentanil treatment had an effect on pain ratings. Therefore, all retrospective pain ratings from both stimulus types and time points were pooled. A comparison between the NaCl and both Remi groups showed a significant pain reduction during remifentanil treatment (visual analog scale [VAS] difference: 15.75, $t_{76}$ = 5.56, $p_{corr}<0.001$, Cohen's d = 1.06, *Figure 1A*). In participants that received remifentanil, pain ratings during the treatment phase were decreased by 30.3% compared to the baseline phase while pain ratings decreased by 1.8% in participants that received saline. More than 92% of participants who received remifentanil reported less pain during the treatment phase compared to the baseline phase while this was only true for 64% in the saline group. To test if the instruction about the certainty of receiving an analgesic drug had an effect on remifentanil-induced analgesia, we compared pain reduction in both Remi groups, which revealed no significant difference (VAS difference: 0.18, $t_{51}$ = 0.06, $p_{corr}$ = 1, Cohen's d = 0.01, *Figure 1B*). This pattern was furthermore coherent across all four retrospective ratings and online ratings (*Figure 1C*, *Figure 1—figure supplement 1*, *Supplementary file 1*). In a next step, we elaborated if this nonsignificant result between both Remi groups indicates that our expectation manipulation had no effect on pain ratings. We thus performed an equivalence test assuming five VAS points difference as the smallest effect size of interest between groups, which yielded no significant equivalence ($t_{51}$ = 1.55, p=0.063), indicating no equivalence between both Remi groups. Treatment beliefs revealed that 100% of participants in the Remi100 group thought they had received remifentanil while 88% in the Remi50 group were sure to have received the drug. In the NaCl group, almost 40% of participants indicated that they had received the drug. To assess expectation effects further, we

performed subgroup analyses based on participants' belief about the treatment they had received. In a first step, we investigated whether pain perception in the NaCl group was influenced by their treatment belief. This effect was not significant (b = 6.45, $t_{21}$ = 1.23, p=0.23). In a second step, we analyzed across NaCl and Remi50 groups whether apart from having received the drug the participant's belief had an influence on pain perception. This analysis revealed a significant effect of treatment (b = 12.01, $t_{45}$ = 3.18, p=0.003) and a nonsignificant effect of belief (b = 6.21, $t_{45}$ = 1.56, p=0.13) on pain perception. Consequently, we pooled both remifentanil groups for all subsequent fMRI analyses (plots showing fMRI results for all three experimental groups can be found in *Figure 1—figure supplement 2*). Since we did not model temporal remifentanil effects in our fMRI analyses, we tested for temporal effects in opioid analgesia across the treatment phase. This analysis revealed no significant difference in pain ratings when comparing pain ratings between sessions and across NaCl and Remi groups ($F_{76,5}$ = 1.88, p=0.097).

## Opioid-related changes in brain and spinal cord

With regard to fMRI analyses, we first identified regions in the brain and spinal cord that showed a parametric response to two different heat intensities (contrast: Stepwise1 > Stepwise2) with the aim to assess whether our imaging approach can sensitively capture pain-related activation in the central pain system. This analysis was restricted to the baseline phase (i.e., without treatment) and revealed neural responses in widespread brain regions such as the parietal and central operculum, insula, putamen, thalamus, and midcingulate cortex (MCC; *Figure 2—figure supplement 1*). With regard to the spinal cord, we observed, among others, a cluster in the left dorsal horn ipsilateral to stimulation site in spinal segment C6 (Figure S5), which corresponds to a location reported in previous spinal fMRI studies investigating pain (*Sprenger et al., 2015*). Results for this contrast across all slices of the spinal volume can be found in *Figure 2—figure supplement 2*.

In the next analysis, we investigated whether remifentanil treatment modulated activity in brain regions associated with pain processing and pain modulation. Brain regions associated with pain processing that showed a significant decrease in activity during remifentanil treatment included the opercular part of inferior frontal gyrus ($xyz_{MNI}$: 56/9/3, $t_{76}$ = 5.95, $p_{corr}$<0.001, *Figure 2A*, contrast: (Remi treatment > baseline) > (NaCl treatment > baseline)), superior part of the circular insula ($xyz_{MNI}$: 33/10/4, $t_{76}$ = 5.80, $p_{corr}$ = 0.001), supramarginal gyrus ($xyz_{MNI}$: 62/–24/24, $t_{76}$ = 4.94, $p_{corr}$ = 0.015), and thalamus ($xyz_{MNI}$: 6/–21/–2, $t_{76}$ = 4.71, $p_{corr}$ = 0.033). Importantly, this analysis accounts for baseline differences between groups and for unspecific time effects (e.g., an opiate-independent increase in activity in the NaCl group). *Figure 2* further shows parameter estimates to illustrate the remifentanil-induced reduction in activity. Conversely, a brain region associated with pain modulation, the anterior cingulate gyrus (ACC) showed a significant increase in activity during remifentanil treatment ($xyz_{MNI}$: 0/20/–9, $t_{76}$ = 4.98, $p_{corr}$= 0.004, *Figure 2B*, contrast: (NaCl treatment > baseline) > (Remi treatment > baseline)). However, parameter estimates indicated that this region was less deactivated under remifentanil treatment compared to saline treatment. A list of all significant clusters can be found in *Supplementary file 1*, and uncorrected results can be found in *Figure 2—figure supplement 3*.

Apart from remifentanil-induced activation differences in the brain, we were further interested in whether the spinal cord showed reduced responses during remifentanil treatment similar to pain-related brain regions using the same contrast ((Remi treatment > baseline) > (NaCl treatment > baseline)). Here, two regions within the spinal cord displayed a significant decrease in activity during remifentanil treatment, including the left dorsal horn ($xyz_{MNI}$: –5/–47/–151, $t_{76}$ = 4.24, $p_{corr}$ = 0.013, *Figure 3*) and a region in (supposedly) layer X ($xyz_{MNI}$: 0/–45/–150, $t_{76}$ = 4.17, $p_{corr}$ = 0.016). Again, this interaction analysis corrects for baseline differences and opiate unrelated time effects. A list of all significant clusters can be found in *Supplementary file 1*, and uncorrected results can be found in *Figure 3—figure supplement 1*.

In the next step, we were interested in whether brain regions associated with pain processing and pain modulation reflected, in addition to group differences, also individual differences. To answer this question, we correlated individual pain ratings with brain activity across all groups by examining differences between baseline and treatment phase. Among pain-related brain regions, activity correlated negatively with perceived analgesia in regions such as the opercular part of inferior frontal gyrus ($xyz_{MNI}$: 57/6/3, $t_{76}$ = 6.95, $p_{corr}$<0.001, *Figure 4A*), supramarginal gyrus ($xyz_{MNI}$: 50/–28/24, $t_{76}$ = 6.31, $p_{corr}$<0.001), thalamus ($xyz_{MNI}$: 16/–14/9, $t_{76}$ = 6.20, $p_{corr}$<0.001), short insular gyri ($xyz_{MNI}$:

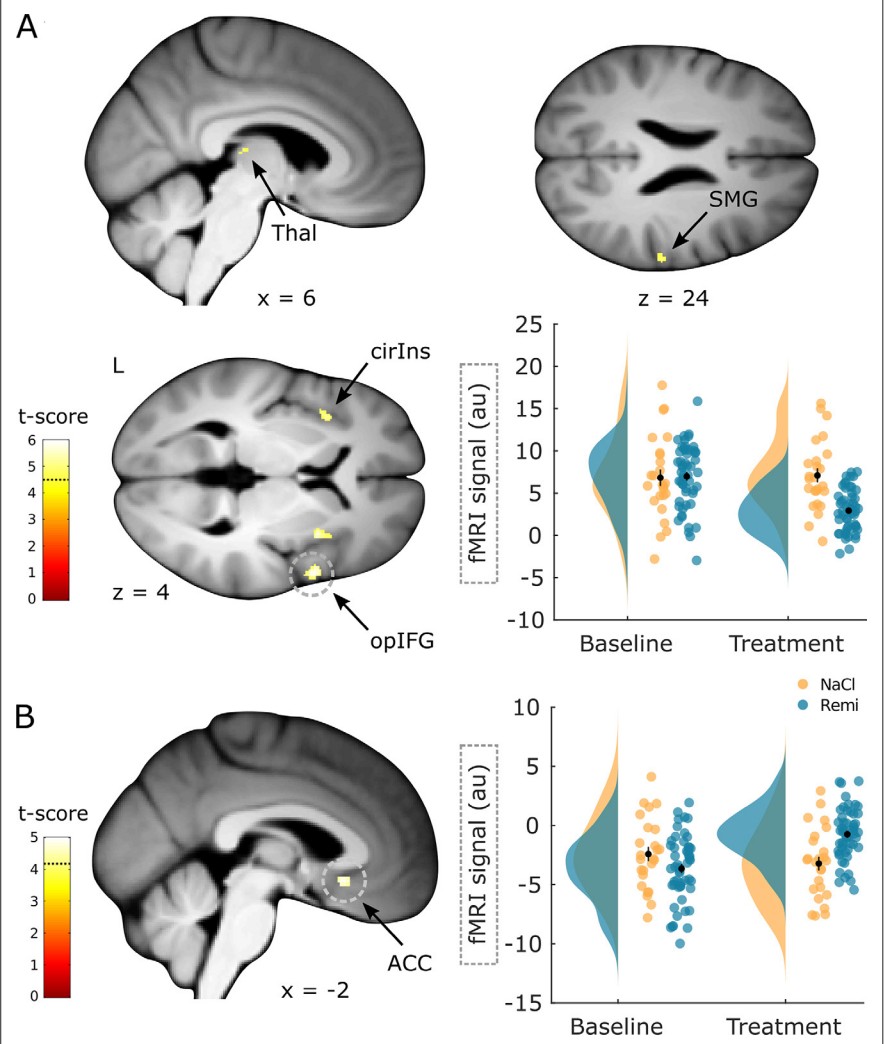

**Figure 2.** Brain regions responding to remifentanil treatment. (**A**) Brain regions that show a significant reduction in activity during remifentanil treatment comprise the thalamus (Thal), supramarginal gyrus (SMG), superior segment of circular insula (cirIns), and opercular part of the inferior frontal gyrus (opIFG). Parameter estimates in the opIFG show reduced activity during the treatment phase in the remifentanil group compared to the NaCl group. (**B**) A brain region that shows a significantly higher activity during remifentanil treatment is the anterior cingulate gyrus (ACC). Parameter estimates show that this region is less deactivated under remifentanil than under saline. Region of interest (ROI)-masked statistical t-maps are overlaid on an average structural T1 image in Montreal Neurological Institute (MNI) template space and the visualization threshold is set to $p_{corr} < 0.05$, which is indicated as a dashed line in the color bar. Dashed circles indicate the cluster from which peak voxel parameter estimates are plotted. Error bars represent standard error of the mean.

The online version of this article includes the following figure supplement(s) for figure 2:

**Figure supplement 1.** Intensity-dependent pain activation.

**Figure supplement 2.** Intensity-dependent pain activation in the spinal cord.

**Figure supplement 3.** Uncorrected interaction results.

–42/0/4, $t_{76}$ = 5.36, $p_{corr}$ = 0.004), and PAG ($xyz_{MNI}$: –8/–33/–8, $t_{76}$ = 4.64, $p_{corr}$ = 0.042). This result implies that stronger analgesia during the treatment phase was associated with a stronger reduction in stimulus-evoked activity in these regions. Time courses in the opercular part of inferior frontal gyrus further show reduced activity during remifentanil treatment and resemble online ratings (*Figure 4B*). In accordance with the brain analyses, we also investigated whether individually perceived analgesia

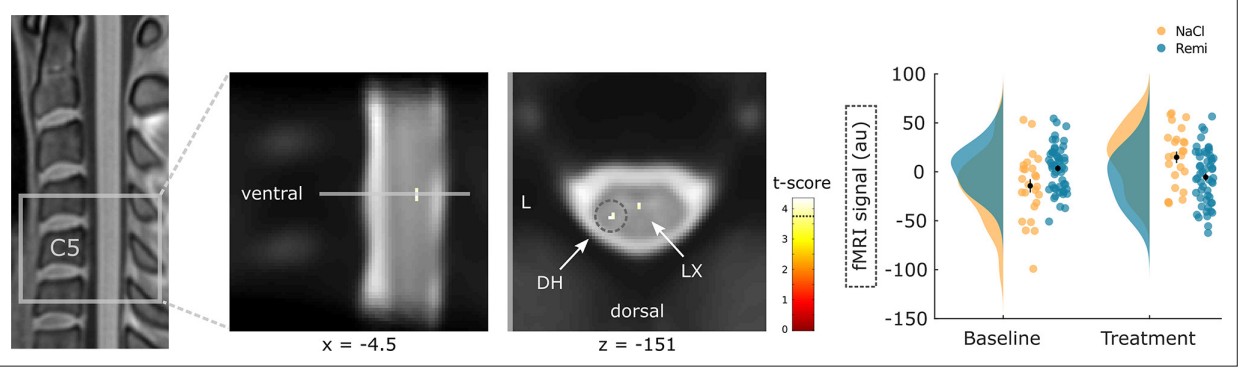

**Figure 3.** Spinal regions responding to remifentanil treatment. Spinal regions that show a significant reduction in activity during remifentanil treatment comprise the left dorsal horn (DH) and layer X (LX). Parameter estimates show that spinal activity in the DH is reduced during remifentanil treatment compared to saline. Region of interest (ROI)-masked statistical t-maps are overlaid on an average mean echo planar imaging (EPI) image in PAM50 template space and the visualization threshold is set to $p_{corr}<0.05$, which is indicated as a dashed line in the color bar. The dashed circle indicates the cluster from which peak voxel parameter estimates are plotted. Error bars represent standard error of the mean.

The online version of this article includes the following figure supplement(s) for figure 3:

**Figure supplement 1.** Uncorrected interaction results in the spinal cord.

correlated negatively with spinal activity. This analysis revealed a cluster in the left dorsal horn that was not significant ($xyz_{MNI}$: −4/–47/–151, $t_{76}$ = 3.44, $p_{corr}$ =0.11).

In contrast, two brain regions in the prefrontal cortex that are associated with pain modulation showed a positive correlation between brain activity and perceived analgesia, namely, the medial orbital sulcus ($xyz_{MNI}$: 8/24/–20, $t_{76}$ = 5.05, $p_{corr}$ = 0.003, *Figure 5A*) and the ACC and sulcus ($xyz_{MNI}$: 6/51/–6, $t_{76}$ = 5.03, $p_{corr}$ = 0.003). Here, stronger perceived analgesia was associated with increased

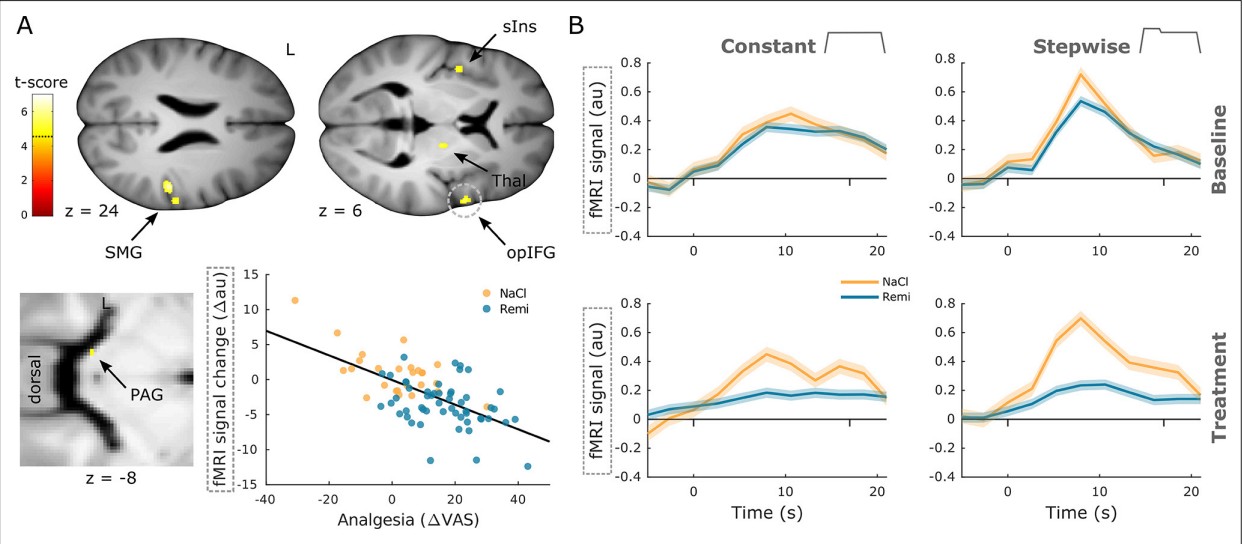

**Figure 4.** Negative correlation between neural activity and analgesia. (**A**) Brain regions that show a significant negative correlation between activity and perceived analgesia comprise the supramarginal gyrus (SMG), short insular gyri (sIns), thalamus (Thal), opercular part of the inferior frontal gyrus (opIFG), and periaqueductal gray (PAG). Parameter estimates in the frontal operculum show that the reduction in brain activity between baseline and treatment phase correlates negatively with perceived analgesia across all groups. (**B**) Time courses extracted from the opIFG correspond to the shape of both stimulus types but are diminished during opioid treatment (dashes denote stimulus onset and offset). Region of interest (ROI)-masked statistical t-maps are overlaid on an average structural T1 image in Montreal Neurological Institute (MNI) template space and the visualization threshold is set to $p_{corr}<0.05$, which is indicated as a dashed line in the color bar. Dashed circles indicate the cluster from which peak voxel parameter estimates are plotted. Shaded areas represent standard error of the mean.

The online version of this article includes the following figure supplement(s) for figure 4:

**Figure supplement 1.** Uncorrected correlation results.

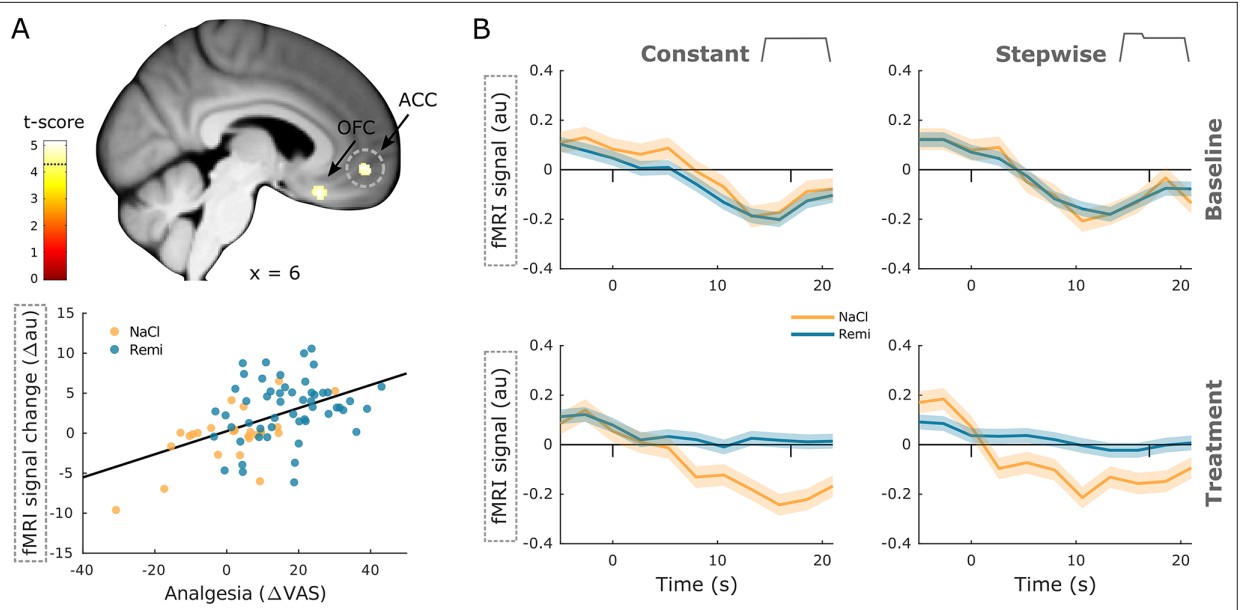

**Figure 5.** Positive correlation between neural activity and analgesia. (**A**) Brain regions that show a significant positive correlation between activity and perceived analgesia across all groups comprise the medial orbital sulcus (OFC) and anterior cingulate gyrus (ACC). Parameter estimates in the ACC show that the reduction in brain activity between baseline and treatment phase correlates positively with perceived analgesia across all groups. (**B**) Time courses extracted from the ACC show a deactivation during noxious stimulation that disappears during opioid treatment (dashes denote stimulus onset and offset). Region of interest (ROI)-masked statistical t-maps are overlaid on an average structural T1 image in Montreal Neurological Institute (MNI) template space and the visualization threshold is set to $p_{corr}<0.05$, which is indicated as a dashed line in the color bar. Dashed circles indicate the cluster from which peak voxel parameter estimates are plotted. Shaded areas represent standard error of the mean.

The online version of this article includes the following figure supplement(s) for figure 5:

**Figure supplement 1.** Overlay interaction and correlation results.

activity during the treatment phase. Time courses in the ACC illustrate that pain-related deactivation is attenuated during remifentanil treatment (**Figure 5B**). A list of all significant clusters can be found in **Supplementary file 1**, and uncorrected results can be found in **Figure 4—figure supplement 1**.

To compare brain regions that reflect group and individual differences, we overlaid results from both interaction and correlation analyses at an uncorrected threshold of p<0.001 (**Figure 5—figure supplement 1**). This overlay revealed adjacent and partly overlapping clusters, indicating that pain-related brain regions represent opioid-induced group differences as well as individual differences in perceived analgesia. One exception was the PAG, which revealed a significant positive correlation but no group differences. Repeating all fMRI analyses for the constant stimulus revealed that many results, especially in the cortex and spinal cord, could be replicated when excluding the high-intensity, temperature-changing stimulus (**Supplementary file 1**).

## Opioid-related changes in the NPS

After identifying individual brain regions that showed remifentanil-induced changes, we examined whether remifentanil also modulated compound activity across several brain regions involved in pain processing. In order to do so, we employed the neurological pain signature (NPS) (**Wager et al., 2013**), which is an fMRI-based marker for pain perception. The NPS assigns different weights to individual voxels based on their contribution to the pain percept and has been shown to correlate with pain perception. We estimated NPS scores pooled across all pain conditions separately for baseline and treatment phases. Results indicated a significant reduction of NPS scores during remifentanil treatment compared to saline treatment (difference score: 14.5 au, $t_{76} = 4.24$, p<0.001, **Figure 6A**), mirroring our behavioral results. A subsequent analysis revealed a negative correlation between NPS scores and perceived analgesia (b = –0.62, p<0.001, **Figure 6B**), indicating that stronger pain relief during the treatment phase was associated with stronger reduction in NPS scores.

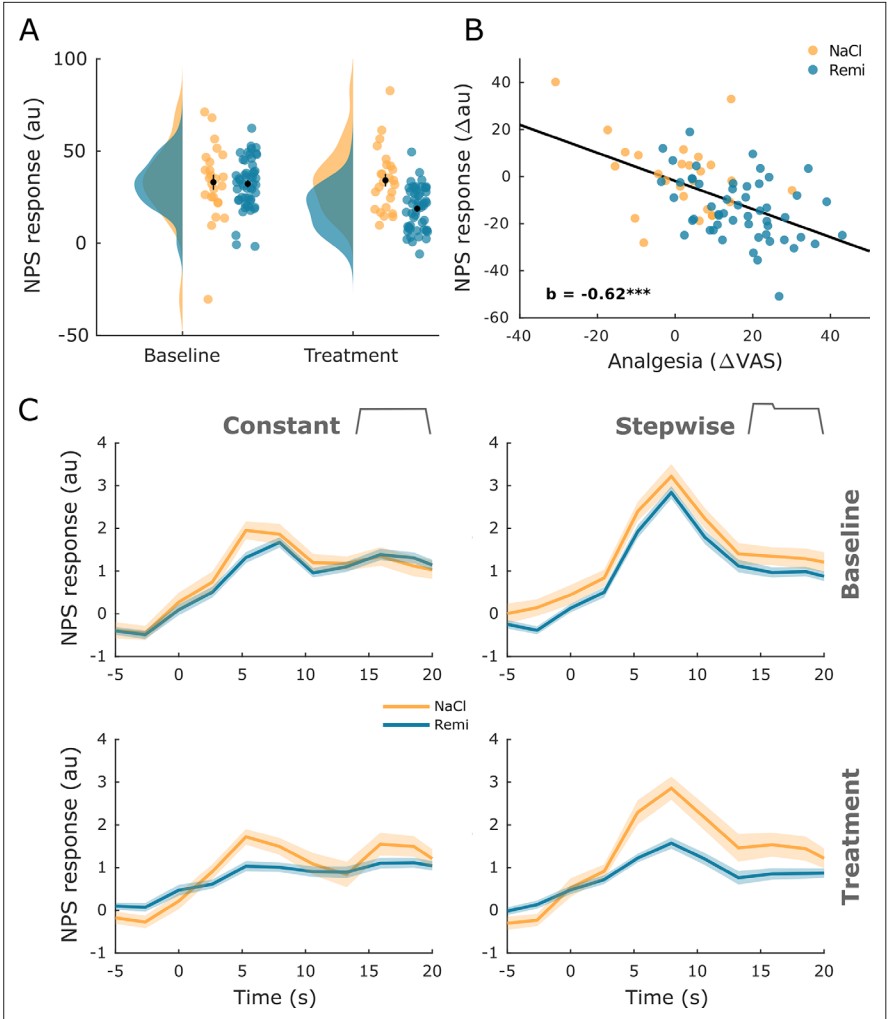

**Figure 6.** Neurological pain signature (NPS) results. (**A**) The NPS scores extracted for every participant and experimental phase show a significant reduction during remifentanil treatment. (**B**) The difference between baseline and treatment NPS scores correlates negatively with perceived analgesia (baseline – treatment). (**C**) The time course of NPS estimates shown for both stimulus types and experimental phases reflects online ratings and shows a reduction of activity in the remifentanil group. Error bars/shaded areas represent standard error of the mean.

Given that our pain stimulus contained some temporal structure, we also tested whether the NPS is capable of revealing the dynamics of pain perception in such stimuli. For this purpose, we estimated a time-resolved NPS score for every TR. This analysis revealed a pattern that closely resembled online ratings, including a reduction during remifentanil treatment (*Figure 6C*). Correlation analyses further showed a significant association between time-resolved NPS responses and behavioral online pain ratings for the baseline phase (pooled across stimuli and groups: $r = 0.57$, $p_{corr} < 0.001$), the NaCl group in the treatment phase (pooled across stimuli: $r = 0.48$, $p_{corr} < 0.001$), and the Remi group in the treatment phase (pooled across stimuli: $r = 0.30$, $p_{corr} < 0.001$).

## Opioid-related changes in functional interactions

Based on contemporary theories of pain resembling a distributed process (*Coghill, 2020*; *Kucyi and Davis, 2017*; *Kucyi and Davis, 2015*; *Lee et al., 2021*) and previous results (*Petrovic et al., 2002*; *Sprenger et al., 2015*; *Tinnermann et al., 2017*), we hypothesized that remifentanil alters the coupling between regions of the modulatory pain system, namely, between the ACC and PAG and between the spinal cord and PAG (*Figure 7A*). More precisely, we expected coupling strength to be associated with individually perceived analgesia similar to previous findings (*Tinnermann et al.,*

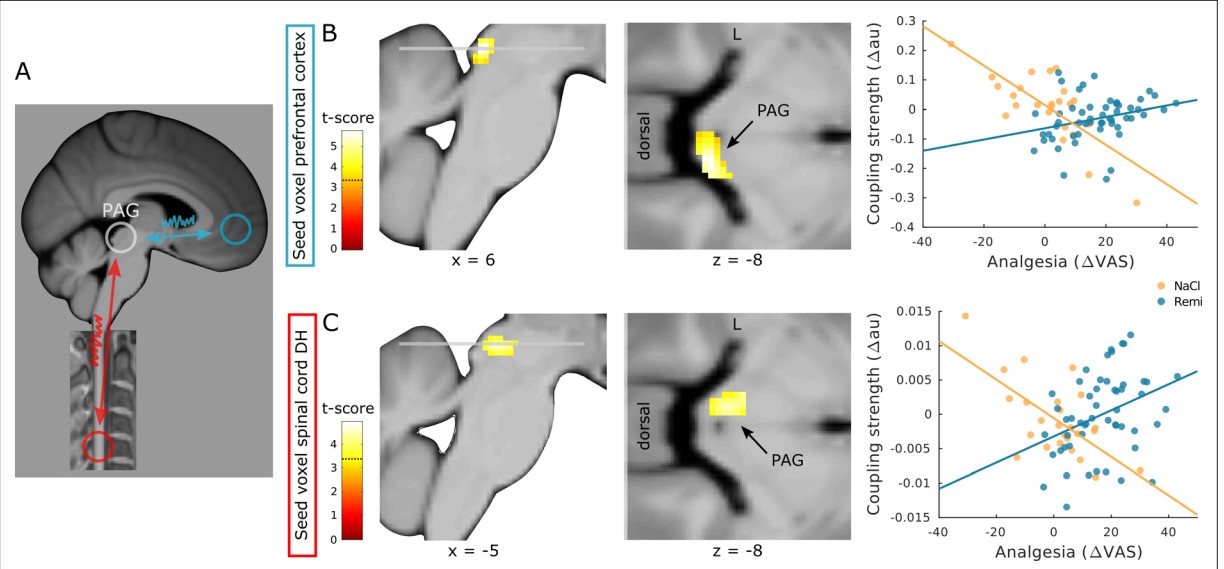

**Figure 7.** Coupling strength analyses between modulatory regions. (**A**) Schematic overview of regions that were examined for opioid-related coupling changes. (**B**) Treatment-related coupling between the anterior cingulate gyrus (ACC) (seed region) and right periaqueductal gray (PAG) correlates more negatively in participants that received saline than in participants that received remifentanil. (**C**) Treatment-related coupling between the spinal cord dorsal horn (seed region) and left PAG correlates more negatively in participants that received saline than in participants that received remifentanil. Region of interest (ROI)-masked statistical t-maps are overlaid on an average structural T1 image in Montreal Neurological Institute (MNI) template space and the visualization threshold is set to $p_{corr}$<0.05, which is indicated as a dashed line in the color bar.

The online version of this article includes the following figure supplement(s) for figure 7:

**Figure supplement 1.** Uncorrected psycho-physiological interaction (PPI) results.

**Figure supplement 2.** Psycho-physiological interaction (PPI) results of periaqueductal gray (PAG) – spinal cord.

*2017*). Moreover, we expected this association between coupling and perceived analgesia to differ between the NaCl and the Remi group. To test this hypothesis, we extracted time courses from regions in the ACC and spinal cord dorsal horn and tested for group differences in pain-related coupling strength between baseline and treatment phase within the PAG. The results of this analysis showed that between the ACC and right PAG, the difference in pain-related coupling strength between baseline and treatment phase (hereafter referred to as treatment-related coupling) correlated more negatively with perceived analgesia in the NaCl group than in the Remi group ($xyz_{MNI}$: 8/–33/–8, $t_{74}$ = 5.73, $p_{corr}$<0.001, *Figure 7B*). A similar pattern was observed between the pain-related cluster in the spinal cord dorsal horn (showing intensity-related activity) and left PAG. Here, the treatment-related coupling also correlated more negatively with perceived analgesia in the NaCl group than in the Remi group ($xyz_{MNI}$: –4/–30/–8, $t_{74}$ = 4.44, $p_{corr}$ = 0.002, *Figure 7C*). These results indicate that stronger perceived analgesia was associated with increased coupling strength during remifentanil treatment while during saline treatment stronger perceived analgesia was associated with decreased coupling strength. Moreover, participants that reported increased pain levels during saline treatment showed increased coupling along regions of the pain modulatory system. A list of all significant clusters can be found in *Supplementary file 1*, and uncorrected results can be found in *Figure 7—figure supplement 1*.

In order to examine whether coupling between the ACC and PAG ultimately targets the spinal cord, we further extracted the time course from the PAG cluster showing coupling with the ACC and verified whether the spinal cord dorsal horn displays treatment-related coupling with the PAG that correlates with perceived analgesia. This analysis revealed a cluster in the dorsal horn that was not significant ($xyz_{MNI}$: –5/–48/–150, $t_{74}$ = 3.48, $p_{corr}$ = 0.108). However, significant group differences in coupling between the NaCl and the Remi group were observed in the left dorsal horn ($xyz_{MNI}$: –3/–47/–150, $t_{76}$ = 3.89, $p_{corr}$ = 0.037, *Figure 7—figure supplement 2*), indicating that coupling strength between the PAG and spinal cord was stronger in the NaCl group than in the Remi group.

## Discussion

In this study, we investigated opioid analgesia along most regions of the central pain system from the spinal cord to the prefrontal cortex. Brain regions associated with pain processing as well as the ipsilateral hemicord within segment C6 showed reduced activity while the ACC showed increased activity during remifentanil treatment. Activity in many of these regions closely followed individual pain ratings. Most importantly, coupling between the ACC, PAG, and spinal cord dorsal horn showed an opposite coupling pattern between the opioid treatment and control group: stronger coupling between these regions was associated with increased opioid analgesia in the remifentanil group while analgesia (e.g., habituation) in the saline group was associated with reduced coupling strength. These results indicate that coupling strength is differentially modulated by individual pain perception and treatment-related modulatory processes, rendering functional interactions along the descending pain system a likely mechanism of opioid analgesia.

### Expectations and opioid analgesia

As expected, we observed that remifentanil treatment significantly reduced pain ratings. However, our expectation manipulation did not show any difference in perceived analgesia between participants that received remifentanil with a 100% certainty compared to participants that received the same drug with a 50% certainty. Previous studies investigating expectation effects in opioid treatments have found that expectations, that is, knowledge about receiving an analgesic drug, can influence reported analgesia (*Atlas et al., 2012*; *Bingel et al., 2011*), indicating that drug effects result from a combination of pharmacological and expectation effects. These studies compared open and hidden treatment conditions where participants either knew that they would receive the drug (100% certainty) or where participants were unaware of the drug treatment (0% certainty). In another study, postsurgical patients received placebo analgesics with either 50% or 100% probability in addition to an active painkiller. Results indicate that patients who received the additional placebo in an open manner requested lower doses of the active painkiller compared to patients that received the additional placebo in a blind manner (*Pollo et al., 2001*). Combining findings from these studies with findings in this study, it is plausible to conclude that a 50% certainty about receiving an active analgesic drug compared to a 100% certainty might not further boost expectations compared to open/hidden paradigms or additional placebo treatments. However, it is also possible that alternative mechanisms account for our behavioral results. The majority of participants in the Remi50 group correctly guessed that they had received the drug, which is in contrast to the study by *Atlas et al., 2012*, where participants were not unblinded in the hidden treatment condition. Since we were mainly interested in the pharmacological effect of remifentanil, we used a higher dose of remifentanil than the previously mentioned study, which might have led to spontaneous unblinding in the Remi50 group due to noticeable drug effects during the treatment phase. Another possibility is that these strong analgesic treatment experiences interfered with expectation effects in a way that expectation effects were negligible in both remifentanil groups. Since we did not include a group that received remifentanil in a hidden context (which is ethically impossible in a between-subjects design), we cannot evaluate whether both Remi groups showed expectation effects on top of the pharmacological drug effect in their pain assessment.

### Opioid effects in the brain

The fMRI data of the brain showed decreased activity in several different regions that are associated with pain processing such as the secondary somatosensory cortex, insula, and thalamus. These findings are in line with the opioid receptor distribution in the brain (*Corder et al., 2018*) and with findings from previous studies that reported similar effects of opioid treatments on BOLD responses (*Atlas et al., 2012*; *Bingel et al., 2011*; *Hansen et al., 2015*; *Wanigasekera et al., 2012*; *Wise et al., 2004*; *Wise et al., 2002*). An important new aspect of our findings is that we did not only find group differences in activity between remifentanil and saline treatment but that activity differences between baseline and treatment phase also correlated with perceived pain changes across all participants irrespective of group, indicating that stronger decreases in brain activity were also associated with stronger analgesia. This result implies that BOLD changes correlate with differences in pain perception irrespective of the cause, leading to altered pain perception since nonspecific effects such as expectations, habituation, or sensitization, as well as opioid analgesia, showed a similar pattern. Furthermore, we found increased activity in the ACC during remifentanil treatment. Similarly to brain

regions involved in pain processing, the ACC displays a high density of opioid receptors (**Corder et al., 2018**). Opioid-related increases in activity have been reported in the orbitofrontal cortex (**Atlas et al., 2012**) and ventromedial prefrontal cortex (vmPFC) (**Bingel et al., 2011**; **Wagner et al., 2007**). Our results further shed light on the activity pattern in this region since the ACC was strongly deactivated during painful stimulation, but during remifentanil treatment this deactivation was attenuated or even absent. Moreover, the more analgesia a participant perceived in the treatment phase, the more attenuation of deactivation was observed in this region. The BOLD time course in the ACC further showed that activity decreased after the onset of painful stimulation and that this decrease was absent during remifentanil treatment. This finding is particularly interesting as pain-related deactivation in the prefrontal cortex has been reported in several imaging studies (**Coghill et al., 1994**; **Kong et al., 2010**; **Porro et al., 1998**). The implication of this deactivation during pain is unclear, but it has been argued that it might be related to task-induced deactivation of the default mode network or to the recruitment of the descending pain system (**Kong et al., 2010**). Interestingly, this deactivation does not seem to scale with pain intensity since the level of deactivation has been found to be similar for low and high painful stimuli (**Kong et al., 2010**). Although we did not test this formally, the time courses in the ACC conformed to this finding because their shape for constant and stepwise stimuli was comparable. However, the magnitude of reduced deactivation during remifentanil treatment did correlate with perceived analgesia in this study. This finding rather supports the hypothesis that prefrontal activity is related to the descending pain system. Since it has been shown in rodents that descending pain modulation is a necessary component of opioid analgesia (**Basbaum et al., 1977**; **Kiefel et al., 1993**), one plausible conclusion is that less deactivation within the ACC during opioid treatment might lead to a stronger recruitment of the descending pain pathway. Other evidence for the role of the prefrontal cortex in the descending pain system comes from studies investigating placebo hypoalgesia. A possible link between opioid and placebo analgesia is that expectations involve endogenous opioids that modulate pain perception (**Amanzio and Benedetti, 1999**; **Eippert et al., 2009a**; **Levine et al., 1979**; **Levine et al., 1978**) similarly to exogenous opioids and that both processes share a common neural mechanism (**Petrovic et al., 2002**). Moreover, gradual response patterns have been described in the prefrontal cortex, showing that stronger placebo-related pain reductions were accompanied by increased neural activity in the rostral ACC (**Geuter et al., 2013**; **Kong et al., 2009**; **Petrovic et al., 2002**). Importantly, this activity pattern in the prefrontal cortex might be linked to increased opioid binding, as has been suggested by a study showing that stronger placebo effects were associated with increased opioid binding in the rostral ACC (**Wager et al., 2007**).

## Opioid effects in the spinal cord

In addition to decreased activity in pain-related brain regions, we further observed remifentanil-induced decreases in BOLD responses in the left dorsal horn and around the central canal. Reduced activity in the left dorsal horn, ipsilateral to the stimulation site, conforms to neurophysiological knowledge about spinal circuits and ascending pain pathways. Peripheral nociceptors synapse onto spinothalamic projection neurons in the ipsilateral, superficial layers of the dorsal horn (layers I and II) and deeper layer V (**D'Mello and Dickenson, 2008**). Superficial layers further display the highest opioid receptor densities within the spinal gray matter (**Faull and Villiger, 1987**), but deeper spinal layers (V, VI, VIII, IX, and X) also express opioid receptors although at lower densities (**Wang et al., 2018**), which could explain why we also found activity in more medial regions around the central canal. This is further in line with studies showing that opioids had an effect on nociceptive spinal processing in deeper dorsal horn layers (**Gogas et al., 1991**; **Presley et al., 1990**). However, it is important to note that the reported opioid-related clusters in the spinal cord were located 3–4 mm more caudally than the pain-related cluster. This finding raises the question whether modulatory effects are directly processed within the ascending pain system or whether modulatory effects are processed in other spinal circuitries within the same spinal segment. Evidence for the former or the latter is rather inconclusive. On the one hand, early work on descending pain modulation has shown that cells that originate in the brainstem and that can inhibit and facilitate spinal nociceptive processing are either directly connected to ascending projection neurons or through one interneuron (**Fields, 2004**). Furthermore, two human fMRI studies that investigated descending pain modulation in the spinal cord have reported reduced spinal activity in a similar location as pain-related activity (**Eippert et al., 2009b**; **Sprenger et al., 2012**), which might suggest that modulatory processes are directly integrated within the ascending

pain pathway. On the other hand, two spinal fMRI studies have reported modulation-related activity in more caudal and medial regions than pain-related activity (*Geuter and Büchel, 2013*; *Tinnermann et al., 2017*), suggesting that additional spinal circuits might be involved in descending pain modulation. However, additional studies, possibly with even higher spatial resolution and improved signal to noise ratio at 7T (*Barry et al., 2016*), would be helpful to further investigate this hypothesis.

## Opioid effects on functional interactions

With regard to coupling along the pain system, we found coupling between the ACC and PAG and between the spinal cord and PAG that correlated with pain ratings. Interestingly, similar to previous work (*Tinnermann et al., 2017*), we found an interaction in correlation between experimental groups. Participants in the NaCl group showed decreased coupling along the descending pathway when reporting less pain during the treatment phase while coupling was increased when participants reported more pain. In contrast, the more analgesia participants experienced in the Remi group during remifentanil treatment the stronger the coupling strength was between the ACC, PAG, and spinal cord. With regard to the brain, these results complement findings from other studies that investigated coupling along the descending pathway. For example, opioid treatment has been shown to increase coupling between the rostral ACC and PAG (*Petrovic et al., 2002*), and, most interestingly, the reported location within the PAG was similar to the cluster found in this study, namely, the right lateral PAG. Similarly to opioid analgesia, increased coupling between the prefrontal cortex and PAG has been reported in placebo hypoalgesia (*Bingel et al., 2006*; *Eippert et al., 2009a*; *Ellingsen et al., 2013*; *Wager et al., 2007*), and some studies further found that on the individual level stronger placebo hypoalgesia was associated with stronger coupling between those regions (*Eippert et al., 2009a*). Placebo-related coupling between the prefrontal cortex and PAG has been shown to be opioid-dependent since coupling was abolished under the influence of an opioid antagonist (naloxone) (*Eippert et al., 2009a*). In summary, these results suggest that modulatory effects leading to pain reduction involve increased coupling between the prefrontal cortex and PAG and that opioid and placebo analgesia share a common mechanism with regard to coupling patterns. The results of this study further expand this observed pattern since lower pain perception in the saline group was associated with reduced coupling and therefore shows the opposite coupling pattern, allowing the conclusion that reduced pain is not per se associated with increased coupling but only in the context of pain modulation.

More importantly, using a novel combined brain–spinal cord fMRI protocol, we were able to also assess functional interactions related to opioid analgesia in the brainstem–spinal cord system. With regard to coupling between the spinal cord and PAG, we observed that participants in the saline group who reported more pain during the treatment phase showed increased coupling between the spinal cord and PAG. These results suggest that interindividual differences in pain perception are associated with coupling strength and that subjectively more experienced pain is accompanied by increased coupling between the spinal cord and PAG as previously reported (*Sprenger et al., 2015*). However, this coupling pattern was reversed in the remifentanil group where stronger coupling between the spinal cord and PAG was accompanied by stronger opioid-induced analgesia. Another study that shows how modulatory processes influence coupling strength between the spinal cord and PAG found that enhanced coupling between these regions was associated with increased expectation-induced pain perception (*Tinnermann et al., 2017*). Together, these results suggest that coupling along regions of the descending pain pathway represents a flexible mechanism that is influenced by individual pain levels as well as by modulatory processes that lead to changes in pain perception. Moreover, these results suggest that in contrast to the correlation results between BOLD changes and altered pain perception that were not specific for opioid analgesia, the coupling pattern along regions of the descending pain pathway is more specific for opioid analgesia and can be distinguished from nonspecific effects that similarly alter pain perception such as habituation or sensitization.

It is important to note that coupling analyses based on regression models such as the psychophysiological interaction (PPI) method do not allow strong statements about directionality and causality. However, descending connections between the medial PFC and PAG exist (*An et al., 1998*; *Floyd et al., 2000*) while ascending connections are not known, which makes it probable that the observed coupling resembles a descending pathway. With regard to the spinal cord–PAG connection, ascending and descending pathways exist (*Fields, 2004*). Since spinal cord projection neurons target

rather lateral parts of the PAG (*Keay et al., 1997*) and the study by *Sprenger et al., 2015* found spinal–PAG coupling during noxious stimulation that did not involve modulation, it is possible that the observed coupling between the spinal cord and PAG in this study was rather an ascending than a descending pathway.

## Limitations

This study was only conducted in male participants, which does not allow generalizing results to females. As previously mentioned, opioidergic mechanisms might differ between males and females (*Bodnar and Kest, 2010*; *Loyd et al., 2008*; *Niesters et al., 2010*), rendering it possible that the reported results in this study might look different in females or in a mixed sample. Furthermore, due to the unavailability of remifentanil at the beginning of this study, a full randomization of participants to study groups was not possible, which might have introduced some bias. The unavailability of remifentanil was another reason why the study experimenter could not be blinded during data acquisition. Therefore, we cannot entirely rule out that interactions between experimenter and participants introduced some additional bias. Another limitation of the study was the reduced field of view in the brain, which was chosen to avoid long repetition times at the expense of excluding some regions of the central pain system such as the RVM or primary somatosensory cortex.

## Conclusion

In summary, this study provides evidence that the spinal cord, similar to brain regions, shows reduced activity during remifentanil treatment. More importantly, we were able to show that temporal dynamics in pain perception are governed by functional interactions comprising many regions of the central nervous pain system. In particular, this analysis highlighted that opioid analgesia is related to changes in coupling strength along the modulatory pain system from the medial prefrontal cortex, over the PAG to the spinal cord.

# Materials and methods

## Participants

We enrolled 95 healthy, male participants in this study. Exclusion criteria were any history of neurological and psychiatric diseases, any chronic pain syndrome, acute pain, current medication or drug use, and any MR-incompatible foreign object in the body. Seventeen participants had to be excluded from the sample: technical issues appeared in six participants; three participants were stimulated with a different thermode model and one participant showed an unexpected emotional response. Furthermore, image alignment of spinal fMRI data between sessions failed in six participants, and registration between the functional and structural image could not be achieved in one participant (for more details, see Appendix 1). The final sample consisted of 78 participants (mean age: 25.4 ± 3.8 years). Study participants were assigned to three experimental groups: two groups (N = 26 and 27) received the μ-opioid receptor agonist remifentanil with different instructions concerning the probability of receiving the drug while the third group received saline (N = 25). A more detailed description of participants based on group assignment can be found in *Supplementary file 1*. The sample size was calculated with G*Power, assuming a power of 90%, an alpha level of 5%, and a Cohen's d of 0.92 (including a penalty for small sample sizes), which was based on a previous study (*Pollo et al., 2001*). This resulted in 21 participants per group that would allow the detection of analgesia differences based on different instructions concerning the probability of receiving the real drug. Reasons for restricting the study sample to male participants were (i) safety concerns regarding possible pregnancy, (ii) potential sex differences concerning analgesic efficacy of opioids in humans (*Niesters et al., 2010*), and (iii) potential sex differences in central mechanisms of opioid analgesia in animals (*Bodnar and Kest, 2010*). The study was conducted in accordance with the Declaration of Helsinki and approved by the Ethics Committee of the Medical Council of Hamburg. All participants gave informed written consent before participating in the study and were compensated with 200 euros for their participation.

## Overall structure of the study

The study consisted of a 2-day paradigm in which participants were briefed about the study and the remifentanil application during the first day. During the second day, participants underwent the fMRI experiment. At the first visit, participants gave written consent after being informed about the study. Participants received detailed information about remifentanil, its potential side effects, and appropriate behavior during the fMRI experiment. Moreover, participants confirmed that they did not meet any of the exclusion criteria and underwent a medical examination and an electrocardiogram to rule out cardiac arrhythmia. Participants were further instructed to not consume any food or beverages within 4 hr prior to the experiment. The first visit also included information regarding the probability of receiving the drug based on participants' group assignment, which was part of the expectation manipulation (see below). On the second day, participants were screened for recent drug use and were only allowed to participate in the study if a urine drug test was negative. They further filled out some questionnaires (see below), and an intravenous line was inserted into the right arm. Subsequently, participants were positioned in the MR scanner and prepared for physiological monitoring during scanning (respiration belt, pulse oximeter, blood pressure cuff). All participants received an intravenous saline solution (unrelated to the experimental manipulation) with a baseline rate of ~500 ml/hr. The thermode for heat pain stimulation was attached to the forearm. During the MR shimming procedure for the corticospinal imaging protocol (around 30 min), participants underwent heat pain calibration on the right volar forearm to find optimal temperatures for the heat pain experiment (see below). After shimming and pain calibration, the fMRI experiment was started. Participants experienced 64 heat pain stimuli on the left volar forearm during eight short fMRI sessions that lasted about 7 min each.

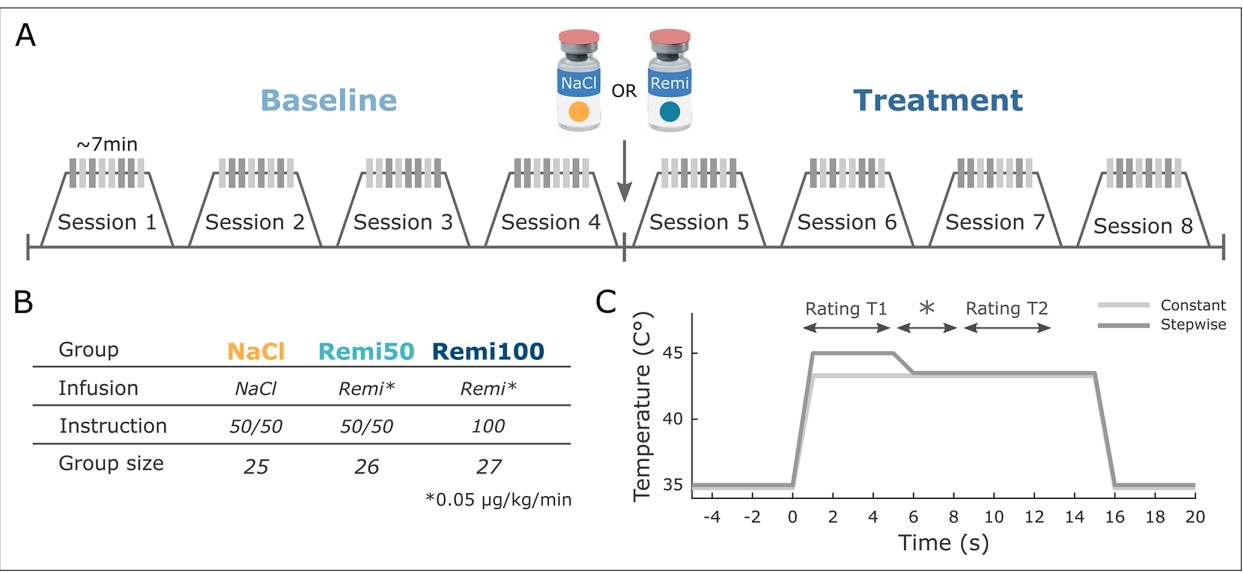

**Figure 8.** Experimental design. (**A**) During the first four functional MRI (fMRI) sessions, all participants received constant and stepwise noxious heat stimuli in a pseudo-randomized order without any treatment (Baseline). Before the fifth session, the infusion pump was started, delivering either saline or remifentanil based on group assignment and the same stimulation protocol was repeated for another four sessions (Treatment). (**B**) Participants were assigned to three different experimental groups. Two groups (Remi50, Remi100) received the opioid receptor agonist remifentanil during the treatment phase while the NaCl group received saline. Participants in the NaCl and Remi50 groups were told that they had a 50% chance of receiving the active drug but remained blind with respect to the actual treatment. The Remi100 group was told that they would receive the drug during the treatment phase, thereby having a 100% probability of receiving the drug. (**C**) Temperature profile of both types of noxious heat stimuli that were applied in the study. The constant stimulus (light gray) was calibrated to be perceived on average as 50 visual analog scale (VAS) and had the same temperature for 15 s. The stepwise stimulus (dark gray) started with a higher temperature (+1.5°C) relative to the calibrated temperature for 5 s (T1) after which the temperature was lowered to the same temperature as the constant stimulus for the remaining 10 s (T2). In the middle of stimulation, a star appeared on the screen for several seconds and participants were instructed to retrospectively rate the painfulness during the 5 s intervals preceding and following the star.

The online version of this article includes the following figure supplement(s) for figure 8:

**Figure supplement 1.** Geometric setup for the corticospinal functional MRI (fMRI) imaging approach.

**Figure supplement 2.** Masks covering regions of interest for small-volume correction.

The first four sessions served as a *baseline phase* without any pharmacological treatment to measure baseline pain levels in every participant. After these four sessions, an infusion of either saline or remifentanil was started and four additional sessions were acquired referred to as *treatment phase* (*Figure 8A*). After the experiment, participants filled out additional questionnaires.

## Experimental groups

Participants were assigned to three different experimental groups (*Figure 8B*). Two groups received the rapid-acting μ-opioid receptor agonist remifentanil during the treatment phase while the third group received saline. The saline group (NaCl) and one remifentanil group (Remi50) received the same instructions. These participants received the treatment in a single-blind manner and were instructed during their first visit that they would receive either saline or remifentanil during the treatment phase, thereby having a probability of 50% for receiving the drug. The other remifentanil group (Remi100) received the treatment in an open manner and was told during their first visit that they would receive remifentanil, thereby having a 100% probability of receiving the drug. Independent of the likelihood of receiving the drug, remifentanil was introduced to all participants as a high-potency opioid with powerful analgesic effects. Due to a worldwide remifentanil shortage during summer 2016 (*Klaus et al., 2018*), a majority of participants belonging to the NaCl group had to be scanned first.

## Remifentanil application

Participants received 0.05 μg/kg/min remifentanil that was continuously delivered through an infusion pump. The remifentanil application was started in the second half of the experiment and lasted for about 35 min. The infusion pump was started directly after session 4 and with preparations for the next session and a thermode change, approximately 5 min passed before the treatment phase, that is, session 5 was started. Participants in the NaCl group received saline via the infusion pump at a rate of 10 ml/hr. There was no significant difference between both remifentanil groups in the amount of remifentanil that was administered (Δremi rate: 0.57 ml/hr; $t_{76} = 0.71$, p=0.48, *Supplementary file 1*). An anesthesiologist monitored participants' vital parameters noninvasively throughout the entire experiment. These parameters included breathing rate, oxygen saturation, heart rate, and blood pressure. Heart rate and respiration rate did not differ significantly between baseline and infusion phase in participants who received remifentanil (Δheart rate: 1.8 beats/min; $t_{76} = 1.55$, p=0.13; Δrespiration rate: 0.8 resp/min; $t_{76} = 1.15$, p=0.26, *Supplementary file 1*). Furthermore, all participants received oxygen (3 l/min) through a nasal cannula as a precautionary measure regarding opioid-related respiratory depression. However, no substantial reduction in oxygen saturation was observed throughout the experiment. In addition, participants had to indicate how they were feeling on a 7-point Likert scale from 'very bad' to 'very good' after every pain rating. These ratings were monitored outside the scanner by a research assistant and an anesthesiologist. Furthermore, participants received a short reminder to breathe calmly through the nose after every trial. After the experiment, participants were invited to eat provided snacks or their own food and to drink sufficient water. Participants were further asked whether they thought they had received remifentanil. In the Remi100 group, 100% of participants thought they had received remifentanil while in the Remi50 group 88% were sure to have received the drug. In the NaCl group, almost 40% of participants indicated that they had received the drug. After the experiment, participants in the NaCl and Remi50 group were unblinded about their treatment.

## Heat stimulus calibration

In order to find a temperature, which participants would rate on average as medium painful, heat pain thresholds (four ramp-ups) were obtained on the right volar forearm using the method of limits, and subsequently, participants underwent a calibration procedure that also familiarized them with different rating schemes that were used in this study. The calibration consisted of 16 heat pain stimuli with varying temperatures, and participants rated the painfulness of each stimulus on a VAS, which consisted of 101 rating steps from 0 to 100. The left anchor of the scale was labeled with 'no pain,' and the right anchor was labeled with 'unbearable pain'.

In some trials during the fMRI experiment, participants were instructed to rate the painfulness of the stimulus on a VAS scale after the stimulation. In other trials, participants rated the painfulness of the stimulus continuously ('online') during the entire stimulation period using the same VAS, and in

some trials, two retrospective ratings were implemented that referred to different time points during the stimulation period and that were signaled by a star (for a detailed description, see below). After this calibration procedure, a sigmoidal function was fitted to the applied temperatures and respective ratings to estimate a temperature for each participant that they would rate on average with 50 VAS rating points. This temperature was rounded to the nearest 0.5°C step for the respective heap pain stimulation protocol. Temperatures above 44.5°C were not accepted due to potential tissue harm considering the 1.5°C increase during half of the stimuli and the stimulus duration of 15 s (see below).

## Experimental design

During the experiment, heat pain stimuli were delivered to the left volar forearm. Prior to the experiment, four skin patches were indicated on the left volar forearm and the thermode was moved to a different skin patch after every second fMRI session. The order of stimulation was randomized between participants. The experiment consisted of two thermal stimulus types (*Figure 8C*). One stimulus had a constant temperature that lasted for 15 s and that was individually calibrated to be perceived as medium painful (50 VAS). The other stimulus had a stepwise temperature profile. Here, the temperature was increased by 1.5°C compared to the constant stimulus temperature during the first 5 s (T1) and then lowered to the medium painful temperature for the remaining 10 s (T2). This stimulus profile was chosen to include aspects of offset analgesia in the study design, which will not be discussed further in this study. In order to separate different levels of painfulness during the stepwise stimulus, a star symbol was shown on the screen for 2.5 s around the temperature decrease and participants were instructed to rate the painfulness during approximately 5 s before the star appeared (T1 rating) and during the 5 s after the star disappeared (T2 rating). The identical rating scheme was also implemented for constant stimuli. Every fMRI session consisted of eight heat pain stimuli, four per stimulus type that were presented in a pseudorandomized order. Two online ratings (one per stimulus type) appeared in a randomized fashion between participants in half of the fMRI sessions, and it was made sure that online rating trials appeared equally distributed among baseline and treatment phase within every participant. In these trials, participants were instructed to continuously rate the painfulness during the entire heat pain stimulation (online rating). The duration of the online rating scale was set to 17 s. To keep the consistency of the overall trial structure, retrospective ratings also appeared after online rating trials. In total, participants received 14 heat stimuli per stimulus type and experimental phase excluding online rating trials.

With regard to the trial design, each trial consisted of an anticipation phase (4 s) that started with a crosshair changing its color from white to red signaling the onset of the next heat stimulus. In online rating trials, additional text appeared during the anticipation phase instructing participants about the continuous rating. During the thermal stimulation, the crosshair remained red. After a thermal stimulation, three rating scales appeared subsequently on the screen and participants had to rate the pain intensity of time points T1 and T2 as well as their mood. A 'breath reminder' appeared for 2 s on the screen followed by the inter-trial interval (ITI) that was indicated by a white crosshair. The formal duration of the ITI was set to 26 ± 3 s to allow the skin to recover, but since rating times were subtracted from ITI duration, the duration of the white cross was shorter and depended on reaction times of the participant. The baseline temperature during anticipation, rating, and ITI was set to 35°C. Thermode rise and fall rates were set to 15°C/s.

## Questionnaires

Participants completed a set of questionnaires during the course of the experiment. The set included German versions of the Beck Depression Inventory, State and Trait Anxiety Inventory, Social Desirability Scale, and Beliefs about Medicines Questionnaire. To assess changes in mood and possible adverse drug effects, the Multidimensional Mood State Questionnaire, a 17-item mood questionnaire, and a 7-item questionnaire assessing typical physical complaints were assessed before and after the fMRI experiment. These questionnaires have been employed by previous pharmacological studies to assess potential drug effects on physical well-being (*Pessiglione et al., 2006*; *Petrovic et al., 2008*; *Sprenger et al., 2012*). All questionnaire scores can be found in *Supplementary file 1*.

## Experimental equipment

Thermal stimulation was realized with a CHEPS Thermode (27 mm surface diameter, Medoc, Ramat Yishai, Israel). Remifentanil/saline application was delivered through an infusion pump (B. Braun, Melsungen, Germany). Stimulus presentation, thermode triggering, and response logging were realized with MatlabR2013a and Psychophysics Toolbox. Apart from monitoring physiological data during scanning with the Expression system (In Vivo, Gainesville, USA), the heart rate and respiration curves were further recorded at 1000 Hz and digitized together with scanner pulses by a CED1401 system and spike2 software (Cambridge Electronic Design) for analysis purposes.

## MRI data acquisition

To investigate opioid analgesia within most regions of the central nervous pain system, we used an improved combined corticospinal fMRI protocol that allows for simultaneous recording of BOLD responses in the brain and cervical spinal cord (*Finsterbusch et al., 2013*), which has been successfully employed in previous studies (*Sprenger et al., 2015*; *Tinnermann et al., 2017*; *Vahdat et al., 2020*). The strength of this approach is that two (brain and spinal cord) subvolumes can be defined with different geometric resolutions and timings. Consequently, optimal parameters for both subvolumes were chosen to obtain an overall optimal image quality for the brain and spinal cord. To be able to measure two subvolumes with different geometries and timings simultaneously, optimal shim parameters and resonance frequencies have to be determined for every subvolume before functional measurements. These parameters are then dynamically updated while recording BOLD responses in the brain or spinal cord, respectively. A full description of the MRI pulse sequence and acquisition strategy can be found elsewhere (*Finsterbusch et al., 2013*).

Imaging data were acquired on a whole-body 3 Tesla TIM Trio system (Siemens Healthcare, Erlangen, Germany) equipped with a 12-channel head coil combined with a 4-channel neck coil (both receive only). Participants were positioned in the scanner avoiding strong extension or flexion of the neck, and foam pads were inserted to minimize movements of the head and neck. The isocenter of the magnet was set to the lower edge of the head coil, corresponding approximately to vertebral level C2/C3 in axial alignment. For the echo planar imaging (EPI) measurements, 40 slices, divided into two subvolumes, were acquired in descending order (*Figure 8—figure supplement 1*). The upper subvolume included 32 slices in the brain, which were initially oriented along the anterior commissure – posterior commissure axis and, if necessary, tilted by maximally 10° to cover the lower part of the pons in the brainstem and most parts of the prefrontal cortex. The lower (spinal) subvolume consisted of 10 slices oriented approximately perpendicular to the spinal cord covering the lower part of the fourth, fifth, and upper part of the sixth cervical vertebral body, and therefore, centered at the spinal segment C6 (*Brooks et al., 2012*). Importantly, MRI acquisition parameters were individually adapted to the optimal setting for each region. Slices in the brain covered a field of view of $220 \times 220$ mm$^2$, with a voxel size of $2.0 \times 2.0 \times 2.0$ mm$^3$, and a gap between slices of 1.0 mm. For the slices in the spinal cord, the field of view was set to $132 \times 132$ mm$^2$ with a voxel size of $1.2 \times 1.2 \times 3.5$ mm$^3$ (without gap between slices) to achieve an adequate signal-to-noise ratio despite realizing a high in-plane resolution. A gradient echo sequence with different timings for the two subvolumes was used to provide optimized parameters for the different resolutions. Parallel imaging using GRAPPA with an acceleration factor of 2 and 48 reference lines was used to reduce echo times and geometric distortions. Thus, echo times of 30 and 34 ms could be achieved for brain and spinal cord slices, respectively, using 7/8 partial Fourier encoding. Acquisition times for the brain and spinal cord slices were 61.2 and 80.3 ms, yielding a repetition time of 2650 ms for the measurement of all 40 slices. Additional fat saturation pulses were applied to both subvolumes to suppress MR signals from fatty tissue.

Since the optimal shim adjustment differs substantially between the brain and spinal cord, a dynamic update of the resonance frequency and linear shims (*Blamire et al., 1996*; *Morrell and Spielman, 1997*) was performed during the EPI measurements. Resonance frequency and linear shim terms were dynamically adapted to the optimized values for the brain and spinal cord subvolume, respectively (*Figure 8—figure supplement 1*). We determined optimal second-order shim values that yielded optimal image quality results for concurrent brain and spinal cord subvolumes. In the spinal subvolume, two saturation pulses were applied anterior and posterior to the spinal target region in a V-shaped configuration to minimize ghosting and inflow artifacts related to pulsatile blood flow in the major cervical vessels (*Figure 8—figure supplement 1*). Furthermore, flow rephasing gradient

pulses were applied in slice direction to minimize signal variations related to pulsatile flow of the cerebrospinal fluid (CSF) (*Giove et al., 2004*). Due to through-slice dephasing (*Cooke et al., 2004*), a compensatory slice-specific gradient momentum was used for the spinal slices (*Finsterbusch et al., 2012*). This 'z-shim' was determined based on prescan acquisitions of the spinal subvolume with 21 equidistant gradient steps applied to all spinal slices and, subsequently, selecting the gradient setting yielding the maximum signal intensity within the spinal cord in each slice. To reduce image noise, only signals from head coil elements were considered for the measurement of brain images, whereas only signals from neck coil elements were considered for spinal images. Because the image reconstruction depends on timing parameters (which differed between brain and spinal cord), optimized reconstructions for brain and spinal cord were performed, respectively (*Finsterbusch et al., 2013*). Subsequently, DICOM files were converted into the NIfTI format and the brain and spinal subvolumes were stored as two separate files. For all analyses regarding the brain, we used the brain-optimized reconstructed images, and for all analyses regarding the spinal cord we used the images reconstructed with the parameters for the spinal subvolume. High-resolution ($1 \times 1 \times 1$ mm$^3$) T1-weighted anatomical images were acquired using a 3D-MPRAGE sequence (sagittal slice orientation, repetition time 2.3 s, echo time 3.45 ms, flip angle 9°, inversion time 1.1 s, and field of view $192 \times 240 \times 320$ mm$^3$). The field of view covered the entire head and neck up to the upper part of the third thoracic vertebra.

## Behavioral data analyses

All data were analyzed with Matlab2017b (The MathWorks). Mean pain ratings were calculated for both stimulus types (constant, stepwise), rating time points (T1, T2), and experimental phases (baseline, treatment). To calculate the overall effect of opioid analgesia, all retrospective pain ratings were pooled and treatment ratings were subtracted from baseline ratings. Furthermore, both remifentanil groups were merged. Statistical significance between remifentanil and saline groups was tested with a two-sample *t*-test (two-tailed). To test if opioid analgesia differed between both remifentanil groups, the difference between pooled baseline and treatment pain ratings for Remi50 and Remi100 was tested using a two-sample *t*-test (two-tailed). Two-sample *t*-tests between remifentanil and saline groups and between Remi50 and Remi100 were repeated for every rating time point and stimulus type to examine whether observed effects were consistent across all stimuli and time points. p-Values were corrected for multiple comparisons (10 tests) using a Bonferroni correction. In addition, we performed a repeated-measures ANOVA with the factors 'Group,' 'Time,' and 'RatingType' using JASP. Post-hoc tests were corrected for multiple comparisons using the Holm method. Results of this analysis can be found in *Supplementary file 1*. Since the difference in perceived analgesia between the Remi50 and Remi100 group was not significant, we further performed an equivalence test assuming independent means, which was realized in the TOSTER R package (*Lakens, 2017*). Mean differences were entered for Remi50 and Remi100, and the lower and upper bound were set to –5 and +5 VAS points, respectively. With regard to online ratings, ratings were averaged across time for both stimulus types and experimental phases. Furthermore, treatment ratings were subtracted from baseline ratings and two-sample *t*-tests (two-tailed) were calculated for comparisons between saline and remifentanil groups and between Remi50 and Remi100 for constant and stepwise stimuli. Bonferroni correction for four comparisons was further implemented. To compare online and retrospective ratings, average pain ratings for both 5 s rating windows during online rating trials in the baseline phase were calculated and compared to retrospective ratings. Results showed a significant correlation between online and retrospective ratings ($r = 0.72$, p<0.001), indicating consistency between both rating types.

For subgroup analyses, we performed two regression analyses, one in the NaCl group and the other across subjects from the NaCl and Remi50 groups to test whether the participants' belief about being in the drug or control group had an influence on pain perception. The first model included 'belief' as a predictor, and the second model included 'belief' and 'treatment' as predictors.

Since the treatment phase started shortly after the remifentanil infusion was started, we tested for temporal effects in opioid analgesia to exclude the possibility that the analgesic effect manifested mainly in later sessions of the treatment phase. Therefore, we performed another repeated-measures ANOVA with the factors 'group' and 'session' in the treatment phase.

## Preprocessing brain data

Brain and spinal data were preprocessed separately, accounting for different requirements of the respective subvolumes. Generally, the first four volumes were discarded to eliminate T1 saturation effects. Brain data were analyzed with SPM12 (Wellcome Trust Centre for Neuroimaging, London, UK) and Matlab2017b, and spinal data were analyzed with SPM12 and the Spinal Cord Toolbox (*De Leener et al., 2017*). For the preprocessing of the brain images, T1-weighted anatomical images were coregistered to the T1-weighted anatomical MNI152 template brain from the VBM toolbox to correct for the low isocenter of the images around vertebrae C2/C3. The same coregistration parameters were applied to all functional images. To remove possible ghosting artifacts outside the brain, EPI images were masked using the ArtRepair toolbox. Subsequently, all EPI images were realigned using rigid-body motion correction with six degrees of freedom. The mean EPI image was then coregistered to the anatomical image and subsequently segmented and spatially normalized to the Montreal Neurological Institute (MNI) standard space using the VBM T1-weighted anatomical template brain and Dartel. For combined acquisitions, normalization of EPI images yields better results than normalization of the corresponding T1-weighted anatomical images (*Finsterbusch et al., 2013*). Next, all functional brain images were warped with individual Dartel flow fields. After performing first-level analyses, contrast images were smoothed with a 6 mm full width at half maximum (FWHM) isotropic Gaussian kernel.

T1-weighted anatomical images were separately segmented and normalized to the same VBM T1-weighted anatomical template brain using Dartel so that T1-weighted anatomical and EPI images were in the same standard MNI space. To create a template brain for illustration purposes, the skull was removed from all normalized T1-weighted anatomical images and a mean image from all participants was generated.

## Preprocessing spinal data

For spinal data preprocessing, the spinal cord was first segmented using a deep learning model. In six participants, another model (deformation model) had to be used to improve segmentation. Subsequently, vertebral bodies were automatically labeled along the spinal cord, which failed in 14 participants and required manual adjustment. T1 images were then normalized to the PAM50 T1 template using a two-step registration procedure based on the segmented spinal cord and the full T1 image. To improve normalization results, parameters such as the smoothing kernel, algorithm type, or metric were individually adjusted. Since the registration between EPI and anatomical images in native space is more robust when using a T2 reference image compared to a T1 image, the PAM50 T2 template image was warped into native space using the T1 reverse warp field resulting from the normalization step.

With regard to functional images, motion correction was the only preprocessing step that was not realized within the Spinal Cord Toolbox because results were more robust using SPM's 3D motion correction with six degrees of freedom than alternative slice-wise approaches. Since motion correction between eight sessions did not yield satisfactory results in many participants using standard procedures, motion correction was performed separately within every session. The resulting session mean EPI images were then registered to the first session mean image using affine transformations, which improved registration results substantially. Parameters such as the metric and gradient step were adjusted across participants to further improve individual registration results. Next, an overall mean EPI image across all sessions was calculated and the spinal cord was segmented from this mean EPI image. For the registration between the mean EPI image and T2 image in native space, a two-step process was implemented using the segmented spinal cord and mean EPI image. Again, registration parameters such as the smoothing kernel, metric, and algorithm were adjusted between participants to improve registration. Finally, the mean EPI image was normalized to the PAM50 T2 template and warp fields from all registration steps were concatenated to reduce interpolation errors and subsequently applied to the functional images. Results of preprocessing steps such as realignment, coregistration, and normalization were carefully revised through visual inspection to ensure that spinal data were correctly registered to the respective template. A special focus for quality control was set on the height of spinal discs and the location of the spinal cord in 3D space. Motion parameters within and between sessions, calculated as root-mean-square translations (*Van Dijk et al., 2012*), are provided in *Figure 8—figure supplement 1*. After first-level analyses, contrast images were smoothed with a 1 ×

1 × 2 mm FWHM Gaussian kernel to facilitate group analysis, but allowing an appropriate anatomical assignment within the spinal cord (*Eippert et al., 2017*; *Tinnermann et al., 2021*).

## First-level analyses

The statistical fMRI data analysis was performed using a general linear model (GLM) separately for the brain and spinal cord subvolumes. The first-level design matrices for both subvolumes included regressors for the experimental conditions 'cue,' 'constant' (17 s), 'stepwise1' (6 s), 'stepwise2' (11 s), as well as regressors for the three different ratings, the breath reminder, and a session constant. The duration of pain-related regressors was slightly increased similar to *Atlas et al., 2012* to account for prolonged heat-induced BOLD responses and for temperature rise/fall times. In experimental runs with online rating trials, the cue and rating period were modeled separately from normal pain trials. Additionally, six motion regressors were included in all first-level analyses and individual volumes were excluded from first-level models if their mean voxel intensity exceeded 3 SDs from the overall mean voxel intensity of the session, thereby excluding scans that displayed imaging artifacts. We further used physiological noise modeling (*Brooks et al., 2008*) based on the RETROICOR method (*Glover et al., 2000*) to include physiological noise regressors that explain signal changes due to cardiac or respiratory processes. An optimized physiological noise model was implemented, which calculates three cardiac and four respiratory harmonics, and interactions between cardiac and respiratory noise (*Harvey et al., 2008*), resulting in 18 regressors per session. These noise regressors were included in all first-level analyses of spinal data and for reasons of consistency also in brain data. To remove low-frequency fluctuations caused by CSF pulsations, we also included CSF noise regressors in both first-level models. For the brain, the average within a CSF mask was calculated for every volume, whereas in the spinal cord, components within the CSF signal were determined via a principal component analysis that explained 90% of signal variance (*Barry et al., 2014*; *Tinnermann et al., 2017*). A temporal high-pass filter with a cutoff period of 128 s was applied to all first-level analyses removing signal drifts and possible remifentanil-induced low-frequency oscillations (*Leppä et al., 2006*). Regressors representing the experimental paradigm were then modeled by convolving boxcar functions for each regressor with a canonical hemodynamic response function (HRF). This was motivated by previous research showing that remifentanil does not alter the shape of the HRF (*Atlas et al., 2012*). After model estimation, we defined contrasts for (A) baseline stepwise1 minus stepwise2 that tested for regions showing a stimulus intensity response that was unrelated to any treatment, (B) baseline minus treatment pain that tested for regions showing differences during the treatment phase, and (C) baseline minus treatment for the constant stimulus as a sensitivity analysis.

In addition, a second 'post-stimulus averaging' model was created using a finite impulse response (FIR) function. This model was only used to plot time courses of BOLD responses during painful stimulation, but not used for statistical inference. The bin width was set to the TR, and 11 bins were modeled for constant and stepwise stimuli separately. Noise regressors were identical to the first GLM approach.

## Second-level analyses

The resulting contrast images from each participant were then raised to random-effects group analyses using a one-sample *t*-test for (A) and a two-sample *t*-test for (B) separately for the brain and spinal cord. Since we did not observe a difference in perceived analgesia between Remi50 and Remi100, we pooled both remifentanil groups for all fMRI analyses. Second-level contrasts of interest were a parametric pain response (stepwise1 > stepwise2) for (A), and an interaction effect of treatment group ((Remi treatment > baseline) > (NaCl treatment > baseline)) for (B). Importantly, this interaction analysis not only tests for group differences but also accounts for baseline differences and unspecific time effects. Perceived analgesia between both experimental phases in terms of VAS rating points (baseline-treatment) was included as a covariate in a separate second-level analysis for (B) to test for correlations between analgesia and opioid-related BOLD responses. Please note that for visualization purposes parameter estimates were sign flipped in correlation plots. All second-level analyses were repeated for the constant stimulus alone (C) to test the sensitivity of our approach given that we pooled both stimulus types in (B).

## NPS analysis

The NPS is a brain-based biomarker for pain that assigns different weights to voxels based on their contribution to predict pain perception (*Wager et al., 2013*). This approach allows examining the compound activity across all voxels in pain-associated pain regions. Therefore, smoothed contrast images for baseline pain and treatment pain were multiplied with the NPS mask to test the hypothesis that BOLD responses in pain-associated brain regions are reduced during opioid treatment. Using a two-sample *t*-test, we tested if this reduction is significant between the NaCl and Remi groups ((NaCl treatment > baseline) < (Remi treatment > baseline)). We furthermore correlated individual (sign flipped) NPS scores with individual analgesia ratings (pain baseline – pain treatment) using a robust regression to test if greater NPS reductions were also associated with greater BOLD reductions. Again, NPS scores were sign flipped for visualization purposes. In a next step, we wanted to investigate if the NPS response shows a similar time course as the behavioral online ratings. We therefore multiplied all 11 FIR contrast images for both stimulus types and experimental groups with the NPS mask and correlated these temporal NPS responses with online pain ratings. Therefore, we pooled all ratings during the baseline phase across all participants and stimulus types and correlated these with the equally pooled NPS responses. For the treatment phase, we correlated ratings and NPS responses separately for the NaCl and Remi groups. Statistical results were corrected for three comparisons using the Bonferroni method.

## Connectivity analyses

To characterize the functional integration between regions of the modulatory pain system, we implemented PPI models with seed regions in the ACC and spinal cord (*Tinnermann et al., 2017*). These seed regions were defined based on results from previously described second-level analyses. Accordingly, we extracted individual time courses from a region of interest (ROI) in the ACC that showed a positive correlation between perceived analgesia and BOLD changes during the treatment phase. The peak voxel from this result was defined as the center of the ROI except for the x-coordinate, which was set to 0 to account for both hemispheres (*Tinnermann et al., 2017*), and the radius of this ROI was set to 8 mm. Time courses within this ROI were extracted from voxels that exceeded the statistical threshold of 0.75 in the relevant interaction contrast (treatment > baseline). With regard to the spinal cord, we extracted individual time courses from three different ROIs since we had no hypothesis which of the spinal clusters might preferentially show connectivity with the PAG. Hence, time courses were derived from one spinal cluster that showed pain intensity-related activation and from two clusters that showed reduced activity during remifentanil treatment. The center of these ROIs was set to the peak voxels of the respective clusters, and the radius was set to 2 mm. Time courses within these ROI were extracted from voxels that exceeded the statistical threshold of 0.75 in the relevant interaction contrast (baseline > treatment). Since the estimation of functional connectivity can be influenced by physiological noise (*Birn, 2012*) and movement (*Van Dijk et al., 2012*), the extracted time courses were adjusted for variance explained by nuisance regressors such as the mean, RETROICOR, and movement. The four PPI models, one with the time course derived from the ACC and three with time courses derived from the spinal cord, included three regressors, one for the psychological variable (pain), one for the time course, and one for the interaction (time course × pain). Nuisance regressors included in first-level GLMs were also included in the PPI models. The contrast for the psychological variable accounted for differences between baseline and treatment phase (baseline treatment), thereby testing for coupling changes between baseline and treatment phase. Contrast images were generated for the PPI, which tested for pain-related changes in the correlation between regional time courses. A two-sample *t*-test was calculated at the group level including analgesia ratings as a covariate to test if changes in coupling strength correlated with perceived analgesia and if this correlation differed between the NaCl and Remi groups. Here, parameter estimates were also sign flipped for visualization purposes.

An additional PPI analysis was performed within the spinal cord volume using a seed voxel in the PAG to test whether the coupling between the ACC and PAG ultimately targets the spinal cord. Therefore, an ROI in the PAG was defined based on the PAG result from the previously described PPI analysis with a seed voxel in the ACC. The center of this ROI was set to the peak voxel, and the radius was set to 2 mm. Please note that using the same threshold of 0.75 as in the previous PPI analyses resulted in two participants with no surviving voxels for time-course extraction, which is why the

statistical threshold was increased to 0.99 for the relevant contrast (baseline > treatment). The PPI model was identical to the previously described PPI models, except that the model also included CSF regressors to account for CSF fluctuations. Two two-sample *t*-tests were calculated at the group level, one testing for group differences in coupling strength and the other testing for group differences based on reported analgesia.

## Multiple-comparisons correction

Correction for multiple comparisons of the functional imaging data in the brain was performed using masks from Neurosynth using the terms 'pain' and 'vmPFC' (*Liang et al., 2019*). These masks were binarized, smoothed by 1 mm to slightly increase their size, and then adapted to the reduced field of view (*Figure 8—figure supplement 2*). For spinal regions, we used a left dorsal horn mask for spinal segment C6 from the Spinal Cord Toolbox that was similarly smoothed by 1 mm. For PAG analyses, we used a PAG mask that was created by Faull and colleagues and already included smoothing (*Faull et al., 2015*). Anatomical labels for significant peak voxels were retrieved from an atlas implemented in Freesurfer (*Destrieux et al., 2010*). Using a small volume correction based on Gaussian random fields as implemented in SPM, voxels within these masks were considered significant at p<0.05 using a family-wise error (FWE) rate. Additional whole-brain FWE-corrected results are reported in *Supplementary file 1*. Activation maps in main figures show only significant voxels within respective masks. Uncorrected whole-brain/spinal results are shown in supplementary figures using an uncorrected threshold of p<0.001 for brain results and p<0.005 for spinal results. x, y, and z coordinates are reported in MNI standard space.

## Acknowledgements

We thank Nicola Peer for medical assistance and careful monitoring of participants around the opioid administration, Tim Dretzler and Lara Austermann for helping with data acquisition and participant handling, and Julien Cohen-Adad for helpful comments on spinal data processing. We further thank our radiographers for their contribution to collecting the fMRI data.

## Additional information

### Competing interests

Christian Büchel: Senior editor, *eLife*. The other authors declare that no competing interests exist.

### Funding

| Funder | Grant reference number | Author |
|---|---|---|
| German Research Foundation | SFB936/A6 | Alexandra Tinnermann Christian Büchel |
| European Research Council | ERC AdG 20100407 | Christian Sprenger Christian Büchel |
| Max Planck Society | | Alexandra Tinnermann Christian Büchel |
| German Research Foundation | DFG-SP 1668/1 | Christian Sprenger |
| German Research Foundation | DFG-TI 1110/1 | Alexandra Tinnermann |

The funders had no role in study design, data collection and interpretation, or the decision to submit the work for publication.

### Author contributions

Alexandra Tinnermann, Data curation, Formal analysis, Investigation, Methodology, Project administration, Validation, Visualization, Writing - original draft; Christian Sprenger, Conceptualization, Investigation, Methodology, Project administration, Supervision, Writing – review and editing; Christian

Büchel, Conceptualization, Funding acquisition, Methodology, Project administration, Supervision, Writing – review and editing

### Author ORCIDs
Alexandra Tinnermann http://orcid.org/0000-0001-9646-0201
Christian Sprenger http://orcid.org/0000-0002-0307-7383
Christian Büchel http://orcid.org/0000-0003-1965-906X

### Ethics
Human subjects: All participants gave informed written consent. The study was approved by the Ethics board of the Hamburg Medical Association (PV4932).

### Decision letter and Author response
Decision letter https://doi.org/10.7554/eLife.74293.sa1
Author response https://doi.org/10.7554/eLife.74293.sa2

## Additional files

### Supplementary files
• Supplementary file 1. Supplementary tables 1a-h.
• Transparent reporting form

### Data availability
Source Data files (contrast images, pain ratings) have been provided for Figures 1 to 7 on Dryad https://doi.org/10.5061/dryad.bvq83bk9k.

The following dataset was generated:

| Author(s) | Year | Dataset title | Dataset URL | Database and Identifier |
| --- | --- | --- | --- | --- |
| Tinnermann A, Sprenger C, Büchel C | 2021 | Cortico-spinal coupling along descending pain system differentiates between opioid and saline treatment in healthy participants | https://dx.doi.org/10.5061/dryad.bvq83bk9k | Dryad Digital Repository, 10.5061/dryad.bvq83bk9k |

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

## Appendix 1

### Exclusion of participants

With regard to exclusion of six participants for technical reasons, fewer sessions were recorded in three participants due to either scanner or participant issues; one participant received 1/60 of the intended dosage due to an incorrect programming of the infusion pump; in one participant, the infusion had extravasated, resulting in an undefined dosage of remifentanil that reached the brain; and in another participant, the field of view was centered on the fourth instead of the fifth vertebra. We further did not include participants in our sample that were stimulated with the different thermode model because the quality of pain perception and the timings (e.g., rise time) are not identical between the two thermode models, which is especially relevant for the stepwise stimulus. Furthermore, image alignment of spinal fMRI data between sessions failed in six participants and image registration between the functional (mean EPI) and structural image (T2) could not be achieved in one participant. Visual quality control revealed a local minimum for registration results in all seven participants, even after parameter adjustment.

