## [Editor Report]

Here, the authors used sophisticated methods for combined brain and spinal cord functional MRI. They report on the influence of an intravenous opioid, remifentanil (a potential, very short-acting µ-opioid receptor agonist), on ascending and descending pain processing pathways in healthy subjects. Their detailed analysis strengthens findings from previous human and animal studies and revealed novel changes in connectivity in the descending pathway to the spinal cord.

---

## [Decision Letter]

**Decision letter after peer review:**

Thank you for submitting your article "Cortico-spinal coupling along descending pain system differentiates between opioid and saline treatment in healthy participants" for consideration by *eLife*. Your article has been reviewed by 2 peer reviewers, and the evaluation has been overseen by a Reviewing Editor and Richard Ivry as the Senior Editor. The following individual involved in review of your submission has agreed to reveal their identity: Anthony E Pickering (Reviewer #2).

Essential revisions:

1. The experiment was originally intended to assess the influence of expectation on the analgesic effect of remifentanil. However, this was compromised as the 'blinded' subjects in the remi50 group were able to tell they were being given remifentanil. It is not clear to the reader whether those subjects that were still masked showed any alteration in their pain. For example, 60% of the control NaCl group and 12% of the Remi50 group thought they had saline and vice versa. I would have expected some analysis of these effects to be done on the behavioural responses before there is a merger of the Remi50 and the Remi100 groups.

2. Two different thermal stimuli "continuous and stepwise" are used but the rationale is not well justified (p6). The pattern of the "stepwise" stimulus with an initial short high step and then lower plateau appears designed for a study of offset analgesia. Why was this profile chosen? And given that it is designed to elicit a different magnitude and temporal profile of pain response, which it did (and presumably neural activations), then why was it merged with the lower stimulus for the analysis (p9). How would the findings be changed if only the lower stimulus data was analysed (as a sensitivity analysis).

3. Given both queries it would be important for the reader and reviewer to see the original protocol that was granted ethical approval to identify the objectives of the study and be able to judge for themselves to what degree the analysis has followed their initial plans and to what extent it is exploratory. This is especially relevant as the study was powered to be able to detect a large expectancy effect (effect size of 0.92) which was not realised.

4. The precise timings of the scan blocks after commencing the remifentanil infusion is not clear. However, it would seem likely that there were 4 blocks each lasting 7 minutes that were spread over a 30 minute period after the infusion was started. The pharmacokinetics of remifentanil are well known, and the infusion was at a fixed rate to a calculated delivery target of 0.05mcg/kg/min (clinically a relatively low rate of remifentanil infusion). This would mean that there would be a steady accumulation of drug in the plasma compartment and subsequently in the target sites in the brain that will develop over the course of around 15-20 minutes before steady state is reached (see Minto model for Remifentanil that showed a targeted 50% inhibition EEG would take almost 20 mins to achieve Minto et al., Anesthesiology 1997). So there is likely to have been a temporal relationship between the analgesic effect and the time after starting the infusion. This is never mentioned yet previous studies have gone to some lengths to model the effects of Remi for fMRI studies (Wise 2002, 2004) and have used either pumps with a pK model to rapidly achieve a target effect site concentration or have used plasma sampling. The authors seem to have considered remifentanil in a binary way (absent/present) which will be an oversimplification of the situation. They should mention this limitation or better provide an analysis to reassure that such an effect was not seen even at a behavioural level.

5. Within the brainstem – could the RVM be visualised or S1 in the cortex? These have both been implicated in human imaging studies as being regulated by opioids (Oertel 2008 & Wanigasekera, 2012). If these were not in the "field of view" then this should be explicitly stated as a limitation of the approach. It is also an overstatement to say that the study has investigated changes in the "entire central pain system" (p3 and 3x other places).

6. Please take a good look again at the anatomical labelling and correct text/figures appropriately. I have a concern about Figure 5 and the anatomical labelling – especially for anterior insula that strikes me as mid-insula – and arguably placement of the FO. Mid-insula is often attenuated by remifentanil so it's fine it's this region and not anterior-insula, but the authors need to check. Also, the PAG voxel/label is in my opinion not in the PAG, brainstem yes, but outside the grey area and possibly CSF artefact? Same again in Figure 8 – some voxels are but others I would argue are not.

7. P14 the parametric analysis is interesting showing a spinal activation in the baseline condition. However, what happened to the activity in this specific cluster in the presence of remifentanil ie running the same contrast?

8. P14 – it would help the reader to follow the logic if the contrasts for these analyses were made explicit in the Results section. The contrast for the parametric analysis is within subject and is mentioned in the methods as (stepwise1 > stepwise2). I initially presumed that this meant a comparison between the constant step and the higher stepwise response but after re-reading it became apparent that this is a comparison of the two pain ratings obtained from the just stepwise response which may contain an element of offset analgesia. Did this spinal activation also appear when doing the more obvious parametric contrast between the two pulses ie constant versus stepwise?

9. P15-16 similarly the contrast for the next analysis looking for an effect of remi on brain activity is across subjects (Remi(Treatment> Baseline)>Control(Baseline>Treatment)) which is an interaction analysis and uses data pooled from both of the types of pain stimuli. In the spinal cord this is driven by both an apparent increase in the BOLD parameter estimate in the baseline group as well as a decrease in the remi group (Figure 4). Would a decrease have been seen with just the simpler parametric analysis (ie stepwise1>stepwise2) across conditions? Also was this increase in the baseline group related to their belief in the treatment they received (ie expectation)?

10. The negative correlations between remi effects on BOLD and analgesic effect include data from the control group that did not have remi infusion (Figure 5A). The control group did not show an analgesic effect and we do not know if there was any relation between their pain δ and their belief about whether they were receiving the opioid. Do any of these relationships persist if the analysis is restricted to just the remi group? It appears that the correlation as plotted depend upon the points from the control group that have less analgesia or indeed a pro-nociceptive effect. A similar consideration applies to the prefrontal areas that show a positive correlation (in Figure 6)

11. In the abstract the authors report that the coupling strength predicts the size of the analgesic effect of remifentanil. "Moreover, coupling strength along the descending pain system, i.e. between the medial prefrontal cortex, periaqueductal gray and spinal cord was stronger in participants who reported stronger analgesia during opioid treatment while the reversed pattern was observed in the control group." However, their comparison (shown in Figure 8) is with the control group who had a saline infusion and did not show an analgesic effect as a group effect. As noted earlier, it is never demonstrated whether the expectation that they might be receiving remifentanil produces any significant analgesic effect. Therefore, this statement and the underlying comparison seem unfounded in terms of the interpretation of the stated difference between the groups.

12. The time-courses of extracted bold from FO and the relationship to the pain scores is intriguing (Figure 5b). This likely tracks the nociceptive input. Is a similar pattern seen for the BOLD at a spinal level especially for the cluster showing the relationship with the parametric data (supp Figure 5)?

---

## [Author Response]

Essential revisions:1. The experiment was originally intended to assess the influence of expectation on the analgesic effect of remifentanil. However, this was compromised as the 'blinded' subjects in the remi50 group were able to tell they were being given remifentanil. It is not clear to the reader whether those subjects that were still masked showed any alteration in their pain. For example, 60% of the control NaCl group and 12% of the Remi50 group thought they had saline and vice versa. I would have expected some analysis of these effects to be done on the behavioural responses before there is a merger of the Remi50 and the Remi100 groups.

We thank the reviewers for this comment. We now provide additional analyses looking at how expectations about having received the drug influence pain perception. In a first step, we tested whether the participant’s belief in the NaCl group influenced their pain perception. This analysis revealed a non-significant effect (b = 6.45, t_21_ = 1.23, p = 0.23). In a next step, we investigated whether apart from having received the drug the participant’s belief across NaCl and Remi50 groups had an impact on pain perception. This analysis revealed a significant effect of drug (b = 12.01, t_45_ = 3.18, p = 0.003) but no significant effect for belief (b = 6.21, t_45_ = 1.56, p = 0.13). We have added these results to the methods and Results section:

“For subgroup analyses, we performed two regression analyses, one in the NaCl group and a second one across subjects from the NaCl and the Remi50 group to test whether the participants’ belief about being in the drug or control group had an influence on pain perception. The first model included ‘belief’ as a predictor and the second model included ‘belief’ and ‘treatment’ as predictors.

To assess expectation effects further, we performed subgroup analyses based on participants’ belief about the treatment they had received. In a first step, we investigated whether pain perception in the NaCl group was influenced by their treatment belief. This effect was not significant (b = 6.45, t_21_ = 1.23, p = 0.23). In a second step, we analyzed across NaCl and Remi50 groups whether apart from having received the drug the participant’s belief had an influence on pain perception. This analysis revealed a significant effect of treatment (b = 12.01, t_45_ = 3.18, p = 0.003) and a non-significant effect of belief (b = 6.21, t_45_ = 1.56, p = 0.13) on pain perception.”

2. Two different thermal stimuli "continuous and stepwise" are used but the rationale is not well justified (p6). The pattern of the "stepwise" stimulus with an initial short high step and then lower plateau appears designed for a study of offset analgesia. Why was this profile chosen? And given that it is designed to elicit a different magnitude and temporal profile of pain response, which it did (and presumably neural activations), then why was it merged with the lower stimulus for the analysis (p9). How would the findings be changed if only the lower stimulus data was analysed (as a sensitivity analysis).

The reviewer is correct in assuming that the study design included the stepwise stimulus to investigate questions about opioid-related changes in offset analgesia in an exploratory fashion. We further wanted to test whether a shorter version of the traditional offset stimulus (only phases T2 and T3) would be sufficient to elicit offset analgesia. However, we decided against including this aspect of the study in this manuscript for two reasons. First, we thought that the amount of results provided in this manuscript is already quite extensive. Second, the offset analgesia results require an additional analysis of the behavioral results, because we included online ratings and tried a new procedure to assess offset analgesia in retrospective ratings. We included a sentence in the methods section to explain why we chose this stimulus type:

“This stimulus profile was chosen to include aspects of offset analgesia in the study design which will not be discussed further in this manuscript.”

We decided to combine stimulus types, mainly to increase statistical power by including twice as many trials in our analyses. Studies have shown that apart from sample sizes the number of trials is an important factor in robust and reproducible fMRI results (Nee, 2019) and we consider this to be especially important with respect to spinal cord fMRI.

However, we followed this reviewer's suggestion and performed a sensitivity analysis and estimated all contrasts for the constant stimulus only and summarized the results in a table that can now be found in the Supplementary File 1 (table 1d). Results indicate that most results are robust and still exist when only considering half the stimuli although some peak voxels changed minimally, and some clusters failed significance. We included a sentence in the methods section regarding this sensitivity analysis and report the results at the end of the second-level Results section:

“All second-level analyses were repeated for the constant stimulus alone (C) to test for the sensitivity of our approach given that we pooled both stimulus types in (B).

Repeating all fMRI analyses for the constant stimulus revealed that many results, especially in the cortex and spinal cord, could be replicated when excluding the high-intensity, temperature-changing stimulus (Supplementary File 1).”

3. Given both queries it would be important for the reader and reviewer to see the original protocol that was granted ethical approval to identify the objectives of the study and be able to judge for themselves to what degree the analysis has followed their initial plans and to what extent it is exploratory. This is especially relevant as the study was powered to be able to detect a large expectancy effect (effect size of 0.92) which was not realised.

We thank the reviewer for this suggestion. With regard to planned analyses and hypotheses, our ethics board does not require precise analysis details which is why this part is rather vague. Furthermore, the proposal is written in German (translated):

“In experiment 1, using the easily controllable drug remifentanil as an example, we want to show how opioid analgesics influence the dynamic interplay within the central pain network. Particular emphasis will be placed on corticospinal interactions. In addition, the study design will allow investigating to what extent the expectation to receive the drug plays a role.”

4. The precise timings of the scan blocks after commencing the remifentanil infusion is not clear. However, it would seem likely that there were 4 blocks each lasting 7 minutes that were spread over a 30 minute period after the infusion was started. The pharmacokinetics of remifentanil are well known, and the infusion was at a fixed rate to a calculated delivery target of 0.05mcg/kg/min (clinically a relatively low rate of remifentanil infusion). This would mean that there would be a steady accumulation of drug in the plasma compartment and subsequently in the target sites in the brain that will develop over the course of around 15-20 minutes before steady state is reached (see Minto model for Remifentanil that showed a targeted 50% inhibition EEG would take almost 20 mins to achieve Minto et al., Anesthesiology 1997). So there is likely to have been a temporal relationship between the analgesic effect and the time after starting the infusion. This is never mentioned yet previous studies have gone to some lengths to model the effects of Remi for fMRI studies (Wise 2002, 2004) and have used either pumps with a pK model to rapidly achieve a target effect site concentration or have used plasma sampling. The authors seem to have considered remifentanil in a binary way (absent/present) which will be an oversimplification of the situation. They should mention this limitation or better provide an analysis to reassure that such an effect was not seen even at a behavioural level.

We thank the reviewers for this comment. We indeed started the infusion after session 4 and waited around 5 minutes (time it took to communicate with participants and change the thermode on the arm) before we started with session 5. We added a sentence in the methods section to clarify that fact:

“The infusion pump was started directly after session 4 and with preparations for the next session and a thermode change, approximately 5 minutes passed before the treatment phase, i.e. session 5 started.”

With regard to modeling remifentanil concentration in the brain, we agree with the reviewers that we used a simple binary model for remifentanil and we are aware that other studies have carefully modeled temporal effects of remifentanil concentration for functional studies. However, we do not think that our approach compromises our results since our analyses are based on session averages where we compare the average of 4 sessions during remifentanil treatment with the average of 4 sessions during the baseline phase meaning that an increasing target site concentration, i.e. decreasing BOLD responses over time should only have a minor impact and is unlikely to generate false positives. In addition, it has been shown that the plasma concentration of remifentanil increases substantially within a few minutes (although we agree that a steady state is reached later) and neural activity (as assessed with EEG) does not seem to show strong temporal effects once plasma concentration has reached a certain level (Egan et al., 1996).

However, to exclude the possibility of strong temporal effects in our data set, we tested behaviorally whether pain ratings decreased during the treatment phase between sessions and groups. The temporal rating data are now shown in Figure 1—figure supplement 1 and the results of a repeated-measures ANOVA revealed no significant interaction between sessions and groups (F_76,5_ = 1.88, p = 0.097), indicating that most of the remifentanil-induced analgesic effect occurs between session 4 and 5 and not between later sessions.

We now also mention this ANOVA in the manuscript:

“Since the treatment phase started shortly after the remifentanil infusion was started, we tested for temporal effects in opioid analgesia to exclude the possibility that the analgesic effect manifested mainly in later sessions of the treatment phase. Therefore, we performed another repeated-measures ANOVA with the factors ‘Group’ and ‘Session’ in the treatment phase.

Since we did not model temporal remifentanil effects in our fMRI analyses, we tested for temporal effects in opioid analgesia across the treatment phase. This analysis revealed no significant difference in pain ratings when comparing pain ratings between sessions and across NaCl and Remi groups (F_76,5_ = 1.88, p = 0.097).”

5. Within the brainstem – could the RVM be visualised or S1 in the cortex? These have both been implicated in human imaging studies as being regulated by opioids (Oertel 2008 & Wanigasekera, 2012). If these were not in the "field of view" then this should be explicitly stated as a limitation of the approach. It is also an overstatement to say that the study has investigated changes in the "entire central pain system" (p3 and 3x other places).

We thank the reviewers for this comment. As can be seen in Figure 8—figure supplement 1, unfortunately, the RVM or S1 were not in our field of view. This restricted field of view was necessary to keep the TR within an acceptable range. Therefore, we changed the term “entire central pain system” to “many/most regions of the central pain system” and included a sentence in the limitations section:

“Another limitation of the study was the reduced field of view in the brain which was chosen to avoid long repetition times at the expense of excluding some regions of the central pain system such as the RVM or the primary somatosensory cortex.”

6. Please take a good look again at the anatomical labelling and correct text/figures appropriately. I have a concern about Figure 5 and the anatomical labelling – especially for anterior insula that strikes me as mid-insula – and arguably placement of the FO. Mid-insula is often attenuated by remifentanil so it's fine it's this region and not anterior-insula, but the authors need to check. Also, the PAG voxel/label is in my opinion not in the PAG, brainstem yes, but outside the grey area and possibly CSF artefact? Same again in Figure 8 – some voxels are but others I would argue are not.

We thank the reviewers for this comment and carefully checked all anatomical labels. Concerning the insula region in Figure 4 (previously Figure 5), the atlas (Neuromorphometrics, http://www.neuromorphometrics.com) integrated in SPM12 still labels this region as anterior insula. Exploring the surroundings of the coordinate, it seems that the peak voxel is located at the border between anterior and posterior insula. As far as we are aware, common brain atlases often do not divide the insular cortex in more than two regions (anterior and posterior). For that reason, we now used the Destrieux atlas (Destrieux et al., 2010), implemented in Freesurfer, to get more precise anatomical labels for our clusters. The insular region in question is now labeled “short insular gyri” and with regard to the cluster labeled as frontal operculum, this has now been changed to “opercular part of inferior frontal gyrus”. We added a sentence to the methods section to explain the origin of all anatomical labels:

“Anatomical labels for significant peak voxels were retrieved from an atlas implemented in Freesurfer (Destrieux et al., 2010).”

With regard to the PAG, we agree that the voxels that survive multiple comparisons correction appear to be located at the border of the PAG. However, the peak voxel of this cluster is still within the PAG mask (provided by Faull et al., 2015) that we used for small-volume correction in all our PPI analyses. Moreover, the uncorrected maps (Figure 4—figure supplement 1) show that the cluster further extends within the PAG.

7. P14 the parametric analysis is interesting showing a spinal activation in the baseline condition. However, what happened to the activity in this specific cluster in the presence of remifentanil ie running the same contrast?

Applying this parametric contrast for the remifentanil treatment, the cluster is located more caudally and centrally (xyz (mm): -2/-47/-150) compared to the cluster found in the baseline condition (-5/-48/-147). However, this cluster’s location is similar to the activation found in the remifentanil interaction contrast. When plotting parameter estimates from the coordinate of the baseline cluster, these data show that there is a decrease in activity for the higher temperature (Stepwise1) during remifentanil treatment while there is no difference for the lower temperature (Stepwise2) between baseline and treatment, probably due to an already low activation for Stepwise2 during the baseline condition, which can explain why this cluster does not appear in the parametric contrast during remifentanil treatment.

8. P14 – it would help the reader to follow the logic if the contrasts for these analyses were made explicit in the Results section. The contrast for the parametric analysis is within subject and is mentioned in the methods as (stepwise1 > stepwise2). I initially presumed that this meant a comparison between the constant step and the higher stepwise response but after re-reading it became apparent that this is a comparison of the two pain ratings obtained from the just stepwise response which may contain an element of offset analgesia. Did this spinal activation also appear when doing the more obvious parametric contrast between the two pulses ie constant versus stepwise?

We thank the reviewer for this comment. The reported first-level model used a single regressor for the constant stimulus, because we did not want to artificially split the constant stimulus into two phases. Hence, with this model, the obvious comparison between Stepwise 1 > Constant1 is not possible. However, to answer this question we ran an additional model in which we employed two regressors for the constant stimulus to be able to only compare the first 5 seconds of both stimuli, as suggested by the reviewer. This analysis confirms the original analysis and shows a cluster that is located in the left dorsal horn (-2/-47/-151) (Author response image 1) although this cluster is located slightly more caudally than the cluster reported in the paper.

**Author response image 1. sa2fig1:** 

9. P15-16 similarly the contrast for the next analysis looking for an effect of remi on brain activity is across subjects (Remi(Treatment> Baseline)>Control(Baseline>Treatment)) which is an interaction analysis and uses data pooled from both of the types of pain stimuli. In the spinal cord this is driven by both an apparent increase in the BOLD parameter estimate in the baseline group as well as a decrease in the remi group (Figure 4). Would a decrease have been seen with just the simpler parametric analysis (ie stepwise1>stepwise2) across conditions? Also was this increase in the baseline group related to their belief in the treatment they received (ie expectation)?

We thank the reviewers for this comment. First of all, it is important to note that we have to investigate an interaction here, i.e. using the saline group as a control. This is necessary, because neuronal responses to painful stimuli over 8 sessions can show unspecific increases (e.g. sensitization) or decreases (e.g. habituation). This can actually be seen in Figure 3 where spinal responses in the saline group show an increase over time e.g. sensitization. This needs to be contrasted to the change in the remi group (i.e. group x time [pre vs post treatment] interaction), where we see a (small) decrease in activation. With respect to the contrast suggested by the reviewer involving stepwise1 and stepwise2, this analysis would reveal voxels that show a differential effect of remifentanil for different levels of pain (i.e. a multiplicative effect). However, we expected similar effects of remifentanil for both levels of pain.

We also investigated whether these activations were modulated by belief: With regard to spinal responses in the saline group, there was no significant correlation between parameter estimates and their belief (r = -0.29, p = 0.17) or perceived pain (r = 0.05, p = 0.82).

10. The negative correlations between remi effects on BOLD and analgesic effect include data from the control group that did not have remi infusion (Figure 5A). The control group did not show an analgesic effect and we do not know if there was any relation between their pain δ and their belief about whether they were receiving the opioid. Do any of these relationships persist if the analysis is restricted to just the remi group? It appears that the correlation as plotted depend upon the points from the control group that have less analgesia or indeed a pro-nociceptive effect. A similar consideration applies to the prefrontal areas that show a positive correlation (in Figure 6)

We thank the reviewers for this remark, as it touches upon a conceptual issue in our paper that we probably have not communicated clearly enough. Figure 4 and 5 clearly show that there is an overall negative (Figure 4) or positive (Figure 5) relationship between changes in pain perception and BOLD responses, i.e. a main effect of pain perception difference. The simple main effects, i.e. restricting the analysis to either the remifentanil or saline group was not significant. However, the important point here is the absence of an interaction (opIFG: F = 1.08, p_uncorr_ = 0.3; ACC: F = 1.79, p_uncorr_ = 0.19). Importantly, not only the remi, but also the saline group shows changes in pain perception when comparing before to after treatment (with a comparable variance: Var remi: 125.28, Var saline 160.75), however in the saline group these changes cannot be evoked by remifentanil. Therefore, responses in these areas show that BOLD responses seem to correlate with differences in pain perception, but are not specific for opioid analgesia (e.g. due to habituation or sensitization). This is in stark contrast to the PPI analysis, where we clearly show that the spinal cord – PAG and vmPFC – PAG coupling is significantly different between groups, i.e. changes in pain perception evoked by remi compared to those due to non-specific effects (e.g. habituation or sensitization) are reflected by clear differences in coupling. We added some sentences to the discussion to better differentiate between main effects and interactions in our correlation analyses:

“This result implies that BOLD changes correlate with differences in pain perception irrespective of the cause leading to altered pain perception since non-specific effects such as expectations, habituation or sensitization as well as opioid analgesia showed a similar pattern.

Moreover, these results suggest that in contrast to the correlation results between BOLD changes and altered pain perception which were not specific for opioid analgesia, the coupling pattern along regions of the descending pain pathway is more specific for opioid analgesia and can be distinguished from non-specific effects that similarly alter pain perception such as habituation or sensitization.”

11. In the abstract the authors report that the coupling strength predicts the size of the analgesic effect of remifentanil. "Moreover, coupling strength along the descending pain system, i.e. between the medial prefrontal cortex, periaqueductal gray and spinal cord was stronger in participants who reported stronger analgesia during opioid treatment while the reversed pattern was observed in the control group." However, their comparison (shown in Figure 8) is with the control group who had a saline infusion and did not show an analgesic effect as a group effect. As noted earlier, it is never demonstrated whether the expectation that they might be receiving remifentanil produces any significant analgesic effect. Therefore, this statement and the underlying comparison seem unfounded in terms of the interpretation of the stated difference between the groups.

We thank the reviewers for this comment and would like to point out that we do not claim or write in the manuscript or abstract that coupling strength predicts analgesic effects due to remifentanil. In our opinion, the sentence from the abstract just describes the correlation result of the connectivity analyses without any interpretation. This interaction in correlation shows that coupling strength increased with increased perceived analgesia during remifentanil treatment while coupling strength decreased with increased perceived analgesia during saline infusion. Differences in pain perception in the saline group probably stem from nonspecific effects as outlined above (e.g. habituation or other nonspecific effects). In addition, these effects might also be present in participants that received remifentanil. Nonetheless, these analyses provide an important result, namely that the direction in which coupling strength is associated with perceived pain levels differs between groups. We also highlight this finding in the discussion to make clear that increased coupling strength is not per se associated with reduced pain perception but that the modulation of coupling strength depends on specific causes that lead to altered pain perception. We now explicitly mention this in the abstract:

“Moreover, coupling strength along the descending pain system, i.e. between the anterior cingulate cortex, periaqueductal gray and spinal cord was stronger in participants who reported stronger analgesia during opioid treatment while participants that received saline showed reduced coupling when experiencing less pain.”

12. The time-courses of extracted bold from FO and the relationship to the pain scores is intriguing (Figure 5b). This likely tracks the nociceptive input. Is a similar pattern seen for the BOLD at a spinal level especially for the cluster showing the relationship with the parametric data (supp Figure 5)?

We also estimated the FIR (i.e. temporal averaging) model on the spinal data but unfortunately, the spinal data seems too noisy to produce reliable parameter estimates for that many, short time bins. In one previous study from our group (Sprenger et al., 2018), time-courses tracking pain levels from the spinal cord were reported, although being quite noisy. However, in that study, BOLD responses were only recorded in the spinal cord which results in less noisy data as compared to combined cortico-spinal imaging (10% less temporal SNR; Finsterbusch et al., 2013). Moreover, with a TR of 740 ms that study acquired about 480 images per condition compared to a TR of 2650 ms and 84 images per condition in this study which also contributes to the fact that time courses in this study were noisier.